# Explainable AI toward understanding the performance of the top three TADPOLE Challenge methods in the forecast of Alzheimer's disease diagnosis

**Monica Hernandez** [ID]*, **Ubaldo Ramon-Julvez, Francisco Ferraz, with the ADNI Consortium[¶]**

Aragon Institute on Engineering Research, University of Zaragoza, Zaragoza, Spain

[¶] Membership of the ADNI Consortium is listed in the Acknowledgments.
* mhg@unizar.es

**Data Availability Statement:** The authors collected data from the ADNI repository (http://adni.loni.usc.edu). Researchers are able to access to these data

## Abstract

The Alzheimer's Disease Prediction Of Longitudinal Evolution (TADPOLE) Challenge is the most comprehensive challenge to date with regard to the number of subjects, considered features, and challenge participants. The initial objective of TADPOLE was the identification of the most predictive data, features, and methods for the progression of subjects at risk of developing Alzheimer's. The challenge was successful in recognizing tree-based ensemble methods such as gradient boosting and random forest as the best methods for the prognosis of the clinical status in Alzheimer's disease (AD). However, the challenge outcome was limited to which combination of data processing and methods exhibits the best accuracy; hence, it is difficult to determine the contribution of the methods to the accuracy. The quantification of feature importance was globally approached by all the challenge participant methods. In addition, TADPOLE provided general answers that focused on improving performance while ignoring important issues such as interpretability. The purpose of this study is to intensively explore the models of the top three TADPOLE Challenge methods in a common framework for fair comparison. In addition, for these models, the most meaningful features for the prognosis of the clinical status of AD are studied and the contribution of each feature to the accuracy of the methods is quantified. We provide plausible explanations as to why the methods achieve such accuracy, and we investigate whether the methods use information coherent with clinical knowledge. Finally, we approach these issues through the analysis of SHapley Additive exPlanations (SHAP) values, a technique that has recently attracted increasing attention in the field of explainable artificial intelligence (XAI).

## 1 Introduction

Alzheimer's disease (AD) involves a progressive deterioration of neuronal structures and brain function, leading to severe cognitive impairment, dementia, and, ultimately, death [1]. The most consistent risk factor for developing AD is advancing age [2, 3]. Owing to the estimated

in the same way as the authors did. The authors do not have special access privileges that others would not have.

**Funding:** 1) MH, URJ Grant number: PID2019-104358RB-I00 Funder: Ministerio de Ciencia e Innovacion URL: https://www.ciencia.gob.es/site-web/;jsessionid= FE84CCA1BA4EAA27EABCADF4007CCF07 The funders had no role in study design, data collection and analysis, decision to publish, or preparation of the manuscript. 2) MHG Grant number: T 64 20R Funder: Gobierno de Aragon URL: https://www.aragon.es/ The funders had no role in study design, data collection and analysis, decision to publish, or preparation of the manuscript.

**Competing interests:** The authors have declared that no competing interests exist.

**Abbreviations:** AD, Alzheimer's Disease; ADAS, Alzheimer's Disease Assessment Scale; ADNI, Alzheimer's Disease Neuroimaging Initiative; APOE4, Apolipoprotein-E4; BAIPETNMRC, Fluorodeoxyglucose Positron Emission Tomography measures; BCA, Balanced Classification Accuracy; bl, baseline; CDR, Clinical Dementia Rating; CN, Cognitive Normal; CSF, Cerebrospinal Fluid; DTI, Diffusion Tensor Imaging; DX, Diagnosis; ECog, Everyday Cognition; FAQ, Functional Activities Questionnaire; FDG, Fluorodeoxyglucose; HCI, Hypometabolic Convergence Index; M, months from baseline; mAUC, multi-class Area Under the receiver operating Curve; MCI, Mild Cognitive Impairment; MMSE, Mini-Mental State Evaluation; MOCA, MOntreal Cognitive Assessment; MRI, Magnetic Resonance Imaging; PET, Positron Emission Tomography; RAVLT, Rey Verbal Learning Test; RF, Random Forest; SHAP, SHapley Additive exPlanations; ST*CV, cortical parcellation volume of FreeSurfer segmentation; ST*SA, surface area of FreeSurfer segmentation; ST*SV, white matter parcellation volume of FreeSurfer segmentation; ST*TA, cortical thickness average of FreeSurfer segmentation; ST*TS, cortical thickness standard deviation of FreeSurfer segmentation; SVM, Support Vector Machines; TADPOLE, Alzheimer's Disease Prediction Of Longitudinal Evolution; UCBERKELEYAV45, ADNI summary table with AV45 standardized uptake value ratios for 18F-flutemetamol PET; UCSFFSL, ADNI summary table with MRI longitudinal biomarkers; UCSFFSX, ADNI summary table with MRI cross-sectional biomarkers; UPENNBIOMK9, ADNI summary with CSF puncture results; XAI, Explainable Artificial Intelligence; XGB, Extreme Gradient Boosting.

growth rate of the global population aged 65 and above in future decades, a considerable increase is expected in the proportion of people suffering from AD.

Clinical research on treatments against AD is approached in two different directions [4, 5]. Some researchers are working toward developing treatments that slow down or reverse the loss of cognitive abilities. Other researchers are working toward developing ways to diagnose the cause of dementia as early as possible in order to design preventive therapies. To date, the development of effective protective or preventive therapies has been hindered by limited knowledge on the causes and mechanisms of the neurodegenerative processes underlying AD. The community agrees on the importance of early diagnosis and accurate prognosis in the success of either the protective or the preventive approach to the treatment of AD [6].

To conduct diagnosis, clinical experts must interpret high-dimensional multi-modal data, including the clinical history of the patients, the outcomes of different cognitive tests, brain scans of different imaging modalities, and mutations of genetic information [7]. Diagnosis is known to be highly subjective. Beach et al. reported a sensitivity range of 70.9 to 87.3 and a specificity range of 44.3 to 70.8 in the identification of probable AD individuals [8]. Postmortem analyses have reported errors in diagnosis of up to 20% of cases [9]. The prognosis of progression to a more severe condition is even a harder problem.

Computational systems for facilitating the diagnosis and prediction of patient condition may enable clinicians to identify the best treatment options in a personalized manner and assess the changes in disease indicators owing to the effect of treatments [10]. Artificial intelligence (AI) is playing an increasingly relevant role with the emergence of computer-aided systems and compelling tools for performing diagnosis and predicting disease evolution [11, 12]. In recent decades, several initiatives have been conducted in the form of challenges to identify the best-performing systems for the diagnosis or prognosis of AD from different types of biomarkers [13–15].

Among them, The Alzheimer's Disease Prediction Of Longitudinal Evolution (TADPOLE) Challenge is probably the most comprehensive challenge to date with regard to the number of subjects, considered features, and challenge participants [15, 16]. TADPOLE deals with the problem of prognosis. The challenge focuses on forecasting the trajectories of three key features, namely clinical status, cognitive decline, and ventricle atrophy, over a five-year period. The initial objective of TADPOLE was the identification of the best-performing data processing, features, and methods for the prediction of the progression of subjects at risk of developing AD. The challenge provided a dataset from the Alzheimer's Disease Neuroimaging Initiative (ADNI) with a complete history of measurements obtained from demographics, cognitive tests, different imaging modalities, and genetics. In addition, it provided a dataset with forecast measurements in real time, which constitutes an actual evaluation set for AD evolution. In the following, we focus on the problem of clinical status prediction.

The results of the TADPOLE Challenge were revealing and identified tree-based ensemble methods such as gradient boosting and random forest in the first positions for the prognosis of the clinical status [17]. Unexpectedly, the highest position for a neural network method was achieved by a long-short term memory network (LSTM), i.e., the ninth position. Feature importance was studied globally, using the outcomes of all the challenge methods. Amyloid beta measured from the cerebrospinal fluid (CSF) puncture, image features from DTI, and APOE status contributed toward the highest accuracy. Meanwhile, image features from tau, amyloid PET, and FDG-PET contributed toward the worst accuracy. The data processing showed considerable variability with differences in record selection, data encoding, augmentation, and imputation of missing values proposed by the challenge participants.

Owing to the challenge design, TADPOLE provided answers limited to identifying which combination of data processing, feature selection, and methods exhibits the best accuracy

metrics. The variability in data preprocessing and feature selection may contribute toward better or worse accuracy of the methods; hence, it is difficult to determine the contribution of the methods to the accuracy. The same problem arises in the assessment of the importance of a given feature in disease diagnosis. In addition, the challenge focused mainly on improving performance while ignoring interpretability issues one more time. It is well acknowledged that the patterns learned using complex and accurate models may not be coherent with clinical knowledge. Therefore, accurate methods should be provided with a set of easily understandable explanations as to why or how the method reaches a specific decision and whether this reasoning is coherent with the knowledge applied in clinical practice.

To obtain deeper insights into the performance of the TADPOLE Challenge methods, it is necessary to perform a comparative assessment of the different models generated by the methods under the same data preprocessing choices. In addition, it would be of interest to study the importance given by each method to the features as well as to establish whether they are consistent with clinical knowledge in order to gain a better understanding of the mechanisms involved in the correct or incorrect diagnosis and predictions. Thus, the purpose of this study is to delve into the following questions using explainable artificial intelligence (XAI) [18–20]:

- Why did the best methods achieve the best accuracy metrics?

- How can we quantify the contribution of each feature toward achieving the best accuracies?

- Which features were the most meaningful for the best methods?

- Do the best methods use information coherent with clinical knowledge?

The answers to these questions are relevant to increasing or decreasing trust in the best-performing TADPOLE Challenge methods for the prognosis of AD toward realizing actionable systems useful in clinical practice.

Our study focuses on the top three methods, namely gradient boosting (*Frog*), random forest (*ThreeDays*), and support vector machine (*EMC-EB*). We built three models trained with balanced versions of the original TADPOLE training set. We performed the same preprocessing of the data for the three models for fair comparison of the methods. As a first approach toward model interpretability in this problem, we studied the importance given by the models to the features by analyzing the system built-in importance, when available, and the SHapley Additive exPlanations (SHAP) values, a recent technique proposed for XAI [20–22]. Both built-in importance and SHAP have been shown to be suitable for quantifying the contribution of a given feature toward the obtained accuracy. In addition, the various ways of representing the information provided by SHAP explainers enabled us to establish whether the models can use feature values that are coherent with clinical knowledge.

Explainable AI has recently started to provide interesting insights into machine learning methods for the diagnosis of AD. Within the TADPOLE Challenge framework, Moore et al. [17] used permutation-based importance in random forests. They found that the diagnosis, participant identifier, and time delay were the most important features. On the basis of these results, the authors argued that the method worked as expected, as the best predictor of future diagnosis should be the last diagnosis, especially for short time horizons, and time should help modulate the confidence in the prevalence of the last diagnosis label. Nguyen et al. [23] used feature ablation to establish the impact of different features on the prediction performance. The authors found that ablating diagnosis resulted in the most significant drop in the evaluation metrics. The drop obtained by ablating the Clinical Dementia Rating-–Sum of Boxes measurement (CDR-SB) was also significant.

Outside the TADPOLE Challenge framework, El–Sappagh et al. [24] recently proposed the use of SHAP values in a random-forest-based multilayer multimodal detection and prediction model. The detection layer performs classification into Alzheimer's disease, mild cognitive impairment, and cognitive normal subjects (AD/MCI/CN). The prediction layer establishes the distinction between stable and progressive MCI subjects (sMCI/pMCI) in a three-year window. Our study differs from El–Sappagh's work in the following aspects. First, our study is conducted within the TADPOLE Challenge framework; therefore, the data used for training, testing, and evaluation are those established by the challenge organizers. El–Sappagh et al. reduced the feature set to a total of 28 baseline features through automatic feature selection and manual expert verification; features outside the TADPOLE Challenge were included among the 28 features. The interpretability study was reduced to random forests. By contrast, we consider a total of 1818 features from both baseline and follow-up visits. We focus on the problem of prognosis, as it is the objective of the TADPOLE Challenge. We present an interpretability study for three methods, including random forests. Both El–Sappagh's study and our study make valuable contributions toward providing a complete understanding of the problem through interesting methods recently proposed in the emerging field of XAI. These systems may be combined with more sophisticated methods for dealing with multi-modal information toward realizing truly actionable explainable systems [25].

The remainder of this paper is organized as follows. Section 2 describes the datasets and data processing methods used in our study. Sections 3 and 4 describe the methods and evaluation metrics considered in our work. Section 5 presents the results of our study. Section 6 provides an analysis of our results in the context of the findings in the state of the art. Finally, Section 7 discusses the most remarkable results of our work and draws some interesting conclusions.

## 2 TADPOLE Challenge dataset: Description and processing

The training and test sets used in this study were built from the longitudinal dataset provided by the TADPOLE Challenge organizers, namely D1 and D2 data (https://tadpole.grand-challenge.org/Data). In addition, the future set D4 was used for evaluation. From D2, the last record from each patient in D4 was reserved as the test data and all the records from the test subjects were removed from the training set candidates to avoid data leakage.

The original sets include 12 734 samples for training, and 219 and 234 samples for testing and evaluation, respectively. We removed from the training set any record from the subjects in the test set. Further, we balanced the training set by a random under-sampling of the over-represented diagnosis groups. The resulting sets include 7 001 samples for training, and 219 and 210 samples for testing and evaluation, respectively. Table 1 lists the number of subjects for the different diagnosis groups in the data-leakage-corrected and balanced training, test, and evaluation datasets.

Table 1. Number of cognitive normal (CN), mild cognitive impairment (MCI), and Alzheimer's disease (AD) individuals in the training, test, and evaluation sets used in this work.

|  | train (corrected, D1-D2) | train (balanced, D1-D2) | test (D2) | eval (D4) |
|---|---|---|---|---|
| CN | 2730 | 2337 | 106 | 86 |
| MCI | 4141 | 2337 | 98 | 92 |
| AD | 2337 | 2337 | 15 | 32 |
| Total | 9208 | 7001 | 219 | 210 |

The clinical criteria for diagnosis in ADNI can be found in [26]. First, memory complaints were recorded on the basis of cognitive exams. The CN subjects had no complaints, whereas the MCI and AD subjects had complaints. Then, the mini-mental state examination (MMSE) results were observed. These results ranged from 24 to 30 for CN and MCI subjects and from 20 to 26 for AD subjects. The clinical dementia rating (CDR) score was 0 for CN subjects, 0.5 for MCI subjects, and 0.5 or 1 with a mandatory requirement of the memory box score for AD subjects. Finally, the Wechsler Memory Scale-Revised (WMS-R) was used with cutoff scores based on education. Subjects with scores coherent with a given group were selected to participate in the ADNI cohort. From these criteria, the TADPOLE Challenge includes `CDRSB_bl`, `CDRSB`, `MMSE_bl`, and `MMSE` as features where `bl` denotes baseline measurements.

The training and test sets were processed using TADPOLE-SHARE code (https://tadpole-share.github.io). The diagnosis column was created from the feature `DXCHANGE` according to TADPOLE-SHARE convention. As a result, the records were categorized into CN, MCI, and AD labels. After this labeling, a considerable number of records (hundreds) have an empty diagnosis. We performed imputation of these labels using nearest-neighbor interpolation based on the patient closest early time points. Seven records with a single visit and no clinical diagnosis were excluded from the original training set, and 24 patients without clinical diagnosis were excluded from the original evaluation set. Our imputation of diagnosis differs from that used in TADPOLE-SHARE, thus obtaining a larger amount of valuable training data. Then, categorical information was replaced by numerical values, and a total of 89 features evidently irrelevant to the diagnosis problem (e.g., update stamp dates) were manually selected and removed, yielding training and test sets with 1818 features. Finally, empty values were imputed using the values of the closest early time points or the mean of the training set data when nearest-neighbor interpolation could not be applied. The same imputation method is provided with TADPOLE-SHARE code.

A complete description of the TADPOLE features with the source table from ADNI is provided at https://github.com/swhustla/pycon2017-alzheimers-hack/blob/master/docs/data_dictionary.md. The features can be divided into clinical history data, cognitive features, Apolipoprotein E4 gene (APOE4), summary anatomical (MRI, DTI) and metabolic features (PET) computed from images, and cerebrospinal fluid (CSF) biomarkers. These features are typically found in the ADNIMERGE table, which contains key ADNI features in a single table. Additional features from the UCSFFSL, UCSFFSX, BAIPETNMRC, UCBERKELEY, DTIROI, and UPENNBIOMK9 tables are included in the list of TADPOLE features. UCSFFSL and UCSFFSX features are obtained from FreeSurfer segmentations of magnetic resonance imaging (MRI) (L denotes the longitudinal while X denotes the cross-sectional value). The BAIPETNMRC features are summaries from positron emission tomography (PET) images. UCBERKELEY are features from UC Berkeley Florbetapir F18-AV-45 PET (AV45) and F-AV1451 Brain Tau PET (AV1451) analysis. DTIROI are measures from diffusion tensor imaging (DTI). Finally, UPENNBIOMK9 are features from CSF puncture results. These are biomarkers of proteomic nature. For many TADPOLE participant methods, the clinical diagnosis is considered as a feature.

## 3 TADPOLE Challenge methods

The exact reproducibility of the best-performing TADPOLE Challenge methods is difficult owing to the lack of both code availability and the scarce details of their implementation. As one of the objectives of this study is to determine the contribution of pure machine learning methods to the accuracy in diagnosis prediction and focus on interpretability, we implemented plain versions of the different systems trained with the underlying machine learning method.

In the following, we describe the methods used in our work as well as the factors that distinguish them from the TADPOLE Challenge participant methods.

## 3.1 Gradient Boosting (XGB)

The winner of the diagnosis forecast in the TADPOLE Challenge was *Frog*, a system built on a gradient boosting machine with XGBoost [27]. Gradient boosting is a model ensemble of individual decision trees that are trained sequentially such that a new tree improves the error of the previous tree ensemble. XGBoost is an optimized distributed gradient boosting library. Its high efficiency is achieved through a parallel tree boosting algorithm that is known to accurately solve data science problems involving billions of examples.

The most relevant hyperparameters of gradient boosting are the learning rate (`learning_rate`), maximum tree depth (`max_depth`), number of trees to fit (`n_estimators`), and $L^2$ regularization weight (`reg_lambda`). We performed hyperparameter selection using a randomized search with the `RandomizedSearchCV` class available in *scikit-learn* with five-fold cross correlation. We found that the default hyperparameters provided a system whose accuracy is close to that of the best-performing model. Therefore, XGBoost has been executed in our study with the default hyperparameter values: `learning_rate` = 0.3, `max_depth` = 6, `n_estimators` = 100, and `reg_lambda` = 1.

In the TADPOLE one-year follow-up paper [16], the XGBoost system of *Frog* used 70 features from the original data. These features were augmented, yielding a total of 420 features. The selected features included clinical diagnosis, cognitive tests (`ADAS-Cog13`, `CDRSB`, `MMSE`, `RAVLT`), MRI, FDG-PET, CSF measurements, and APOE status. However, the exact list of features is not available. Augmentation was performed on the basis of the most recent measurement and time, historical highest and lowest measurements and time, and most recent change in measurement. Missing data were filled automatically by XGBoost through inference based on the reduction of the training loss. Prediction was performed using different forecast windows.

We depart from *Frog* system implementation by using all the available features and the longest available forecast window. Augmentation was not considered in this work. Although XGBoost can automatically fill in the missing data, we decided to conduct the same imputation procedure for all the methods in our study.

## 3.2 Random Forests (RF)

The runner-up of the diagnosis forecast in the TADPOLE Challenge was *ThreeDays*, a random forest (RF) machine. RF is a model ensemble that consists of a large number of decision trees. Each individual tree in the forest performs one class prediction, and the class with the majority of the votes becomes the model prediction. The trees in a forest are uncorrelated models that together produce ensemble predictions that are more accurate than any of the individual predictions. Uncorrelation is the key to a successful ensemble. It is obtained from random selection of different sets of data and different features for each tree.

The most relevant hyperparameters of RF are the method for sampling data points (`bootstrap`), maximum number of levels in the tree (`max_depth`), number of features to be considered at every split (`max_features`), minimum number of samples required at each leaf (`min_samples_leaf`) and to split a node (`min_samples_split`), and number of trees in the forest (`n_estimators`). As with XGB, we performed hyperparameter selection using `RandomizedSearchCV`. We also found that the default hyperparameters provided the same accuracy as those of the best-performing model. Therefore, RF has been executed in our study with the default hyperparameter values: `bootstrap` = True,

```
max_depth = None, max_features = auto, min_samples_leaf = 1, min_
samples_split = 2, n_estimators = 100.
```

In the TADPOLE one-year follow-up paper [16], the *ThreeDays* system was built with a manual selection of 16 features, namely clinical diagnosis, age, months since baseline, gender, race, marital status, cognitive tests (`MMSE`, `CDRSB`, `ADAS11`, `ADAS-Cog13`, `RAVLT` immediate, learning, forgetting, and percent forgetting, `FAQ`), and APOE status. No imputation was performed in the data, as their RF implementation deals with missing data automatically, by finding optimal splits based on existing data. Two RFs were trained, the first for transitions from CN to AD and the second for transitions from MCI to AD. The authors made the following assumptions in the model: CN individuals always evolve to MCI, MCI individuals evolve to AD, and the diagnosis does not change for AD individuals. The model does not consider valid backward situations in diagnosis. A similar method with a single random forest for all the predictions has been described in [17].

Again, we depart from the *ThreeDays* system implementation using all the available features and a single RF for the diagnosis prediction problem. Our RF implementation did not deal with missing data automatically; hence, the missing data were imputed. We do not consider the diagnosis assumptions, as we can find some counterexamples in the data owing to possible corrections in diagnosis.

## 3.3 Support Vector Machines (SVM)

The top three diagnosis forecasts in the TADPOLE Challenge are completed with *EMC-EB*, a support vector machine (SVM). Given the number of features, *n*, SVM aims to find a hyperplane in an *n*-dimensional space that best classifies the training data points. The hyperplane maximizes the distance between the points of both classes (maximum margin). The selection of the maximum margin hyperplane as the classification boundary increases the confidence in the algorithm for obtaining correct classifications in unseen data.

The most relevant hyperparameters of SVM are the kernel, the C-parameter, which is a regularization parameter related to the margin size, and $\gamma$, which is used to set the size of the kernel. In this work, we used a radial basis function kernel (RBF), C was set to 0.5, and $\gamma$ was automatically selected depending on the number of features. These hyperparameter values were selected by the authors of *EMC-EB*.

In the TADPOLE one-year follow-up paper [16], the *EMC-EB* system used 200 features from the original data. The features were selected from those showing the largest change over time in subjects who progressed to AD. The file can be found with TADPOLE-SHARE source code (http://tadpole-share.github.io). The selected features included clinical diagnosis, cognitive tests, FreeSurfer cross-sectional MRI volumes, PET and DTI measures, and CSF features. Missing data were imputed using nearest-neighbor interpolation from the subject's earlier timepoints. When this was not possible, the missing data were imputed from the mean values in the training set.

In this work, we only depart from the *EMC-EB* system implementation by using all the available features.

## 3.4 Benchmark methods

Benchmark methods were provided by the TADPOLE Challenge organizers. The source code was offered to the participants before the deadline as a starting point for making predictions. The evaluation scores obtained using these methods constitute the lower bound of the accuracy expected for the best-performing methods.

**3.4.1 Last visit.** This method uses clinical diagnosis as the only available feature. The prediction is obtained by assigning 100% probability to the last available diagnosis and 0% to the other diagnoses.

**3.4.2 Mixed effects APOE.** This method uses the diagnosis, age, ADAS13, and APOE status as features. A mixed-effects model is built using the APOE status as a covariate. The subject age at each visit is selected as the predictor variable, and the predictions are derived from the ADAS13 forecasts using three Gaussian likelihood models for the CN, MCI, and AD status. Then, likelihoods are converted into probabilities by normalization and, the prediction is computed from these probabilities.

**3.4.3 SVM.** This method uses the clinical diagnosis, age, ADAS13, ventricles, ICV, and APOE status as features. The SVM is implemented with a linear kernel. Missing data are imputed using the average value of the biomarker from past visits of the same subject or the population average if past information is not available. In this work, we also consider an SVM trained with the whole set of 1818 features.

## 3.5 TADPOLE Challenge metrics for evaluation

The metrics proposed in the TADPOLE Challenge for the evaluation of the systems in the problem of clinical status prognosis were the multi-class area under the receiver operating curve (mAUC) and the overall balanced classification accuracy (BCA). The detailed expressions of these metrics can be found in https://tadpole.grand-challenge.org/Performance_Metrics.

The mAUC is an extension of the classical ROC analysis from binary to multi-class problems. Given $AUC(C_i|C_j)$, the AUC for the classification of a class $C_i$ against $C_j$, the overall mAUC is given by

$$mAUC = \frac{2}{C(C-1)} \sum_{i=2}^{C} \sum_{j=1}^{i} \frac{AUC(C_i|C_j) + AUC(C_j|C_i)}{2}, \tag{1}$$

where $C$ is the number of considered classes. Intuitively, the mAUC measures the degree to which the model is capable of correctly distinguishing between classes; a higher degree is desirable.

The overall BCA is given by the mean of the balanced accuracies for each class:

$$BCA = \frac{1}{C} \sum_{i=1}^{C} BCA_i, \tag{2}$$

where

$$BCA_i = \frac{1}{2} \left( \frac{TP}{TP + FN} + \frac{TN}{TN + FP} \right), \tag{3}$$

and TP, FP, TN, and FN represent the number of true positives, false positives, true negatives, and false negatives, respectively, for the class $C_i$. Intuitively, the BCA provides a balanced assessment of true positives/negatives and false positives/negatives obtained by the model.

## 4 Explainable AI

Explainable AI (XAI) is a recent subfield of AI that aims to provide explanations of general machine learning algorithms while focusing on the most complex families of methods that have been traditionally considered as black boxes [18, 20]. The objective of XAI is to leave out the trade-off existing between accuracy and explainability and provide both powerful and

explainable systems with arguments for increasing the confidence in the output of the algorithms.

Explainability technically highlights the relevant parts of a system that contribute toward model accuracy during training or toward a specific correct or incorrect prediction for a particular observation during testing or evaluation [28]. The concept has been recently linked with causability, which is defined as the extent to which an explanation to a user achieves a specified level of effective, efficient, and satisfactory causal understanding in the context of use [29, 30]. Causability provides criteria for measuring and ensuring the quality of explanations. Explainability linked with causability improves causal understanding; thus, AI methods and explanations become trustable and actionable, i.e., the actions that should be taken become clear for decision makers [30]. Thus, explainability is important first step toward fully actionable and interpretable AI systems.

The different techniques in XAI facilitate the establishment of the importance given to the different features by the system, assessing how a particular feature affects model predictions, or which feature values favor or hinder correct or incorrect predictions. All this information can help ensure whether only significant features coherent with current knowledge are used to infer the output for the application of interest. In this work, we focus on the built-in importance scores of interpretable machine learning methods and SHAP.

## 4.1 Built-in importance scores

XGB and RF are ensembles of decision trees; therefore, the models can provide estimates of the importance given to different features. In XGB, the importance measures how valuable each feature is in the construction of the boosted decision trees within the model. The more an attribute is used to make key decisions within the trees, the higher is its relative importance. The importance is measured from the amount by which each feature split point improves the performance measure (gain). Another measure of importance is the cover, which summarizes the second-order gradient (Hessian) of the loss function for all the predictions of a point in a tree. Accordingly, XGB provides five ways for computing feature importance:

1. weight: directly measures the number of times a feature is used to split the data across all trees

2. gain: average gain across all splits in which a feature is used

3. cover: average coverage across all splits in which a feature is used

4. tgain: total gain across all splits in which a feature is used

5. tcover: total coverage across all splits in which a feature is used

The gain and cover are computed from the quotients of the total gain and total cover with the weight. These metrics are usually recommended for quantifying importance. However, it should be noted that the weight value for a feature depends on the range of different values taken by that feature. Therefore, there may be important variables for the model with a low weight because the range of different values is low (e.g., gender). In this situation, the total gain or the total cover value is preferred. This is the case for our study.

In RF, feature importance is computed from the mean and standard deviation of the accumulation of the impurity decrease within each tree (Gini importance). As with the XGB importance metrics, the feature importance metrics here may be misleading for features with a low number of different values. For kernel-based SVM, there is no built-in way of establishing feature importance.

## 4.2 SHapley Additive exPlanations

SHapley Additive exPlanations (SHAP) has attracted increasing attention in the field of XAI (https://shap.readthedocs.io). The core idea of SHAP is to transfer ideas from cooperative game theory to the attribute feature importance of a model output given an input [31]. SHAP values represent the distribution of the contributions toward game success or failure amongst all the players working in cooperation. In the context of explainability of machine learning methods, SHAP values represent the change in each feature in the expected model prediction under conditioning on that feature. The explainability from SHAP is achieved through different representations derived from the SHAP values. In this work, we use the mean absolute values of the SHAP values obtained for each class and the violin plots of the SHAP value impact on the estimation of the probabilities for each class.

S1 Fig in S1 File shows typical examples of the SHAP value representation for a given problem (survival classification in the Titanic dataset). The horizontal bar plot shows the mean absolute SHAP values of the most relevant features for survival classification. The plot shows that gender is the most relevant feature for the correct classification of survivals, followed by the title code of the person and the class. These results are coherent with our knowledge of the Titanic tragedy. Therefore, the underlying machine learning method can generate a model that is aware of the outcome of survival owing to the human decisions made during the disaster.

The violin plot in Fig 1 represents the impact of the feature values on the probability computed for the survival class by the model. The color code indicates the feature values for different test samples, and it is useful to relate whether the high or low probabilities computed by the model are favored by given feature values. The color bar ranges from blue tones for low values to red tones for high values. Thus, the model favors high values of the probability of survival for high values of gender (i.e., female). From the class feature, we can see that the model favors high values of the probability of survival for low values of class (i.e., first class). In our study, we use similar reasoning to establish the relevance of the features in the identification of different classes (AD, MCI, or CN).

For XGB and RF, the SHAP values are obtained from tree explainers, a fast implementation of SHAP for tree-based methods [32]. For SVM, the SHAP values are obtained from kernel explainers, an agnostic algorithm that consequently entails a higher computational cost.

# 5 Results

## 5.1 Performance study

**5.1.1 Performance results.** Table 2 (left) summarizes the performance of the TADPOLE Challenge methods of interest in this work on the evaluation set (D4). Table 2 (right, top) shows the performance of the baseline methods (last Visit, mixed effects, and linear SVM). These results are the same as those reported at https://tadpole.grand-challenge.org/Results and [16]. Table 2 (right, bottom) also summarizes the performance of the baseline methods trained with our data on the test (D2) and evaluation (D4) sets. These results were obtained by integrating the TADPOLE-SHARE code (https://tadpole-share.github.io) into our code. The results obtained on D4 were close to the TADPOLE Challenge results. Therefore, our choice of data preprocessing provides a similar performance on the benchmark methods.

Table 3 summarizes the performance of the methods considered in this work on the test (D2) and evaluation (D4) sets. We performed an ablation study on feature set selection. Therefore, the experiments were conducted by removing different sets of features, namely, the diagnosis (-D), diagnosis and cognitive features of TADPOLE used in ADNI to establish diagnosis

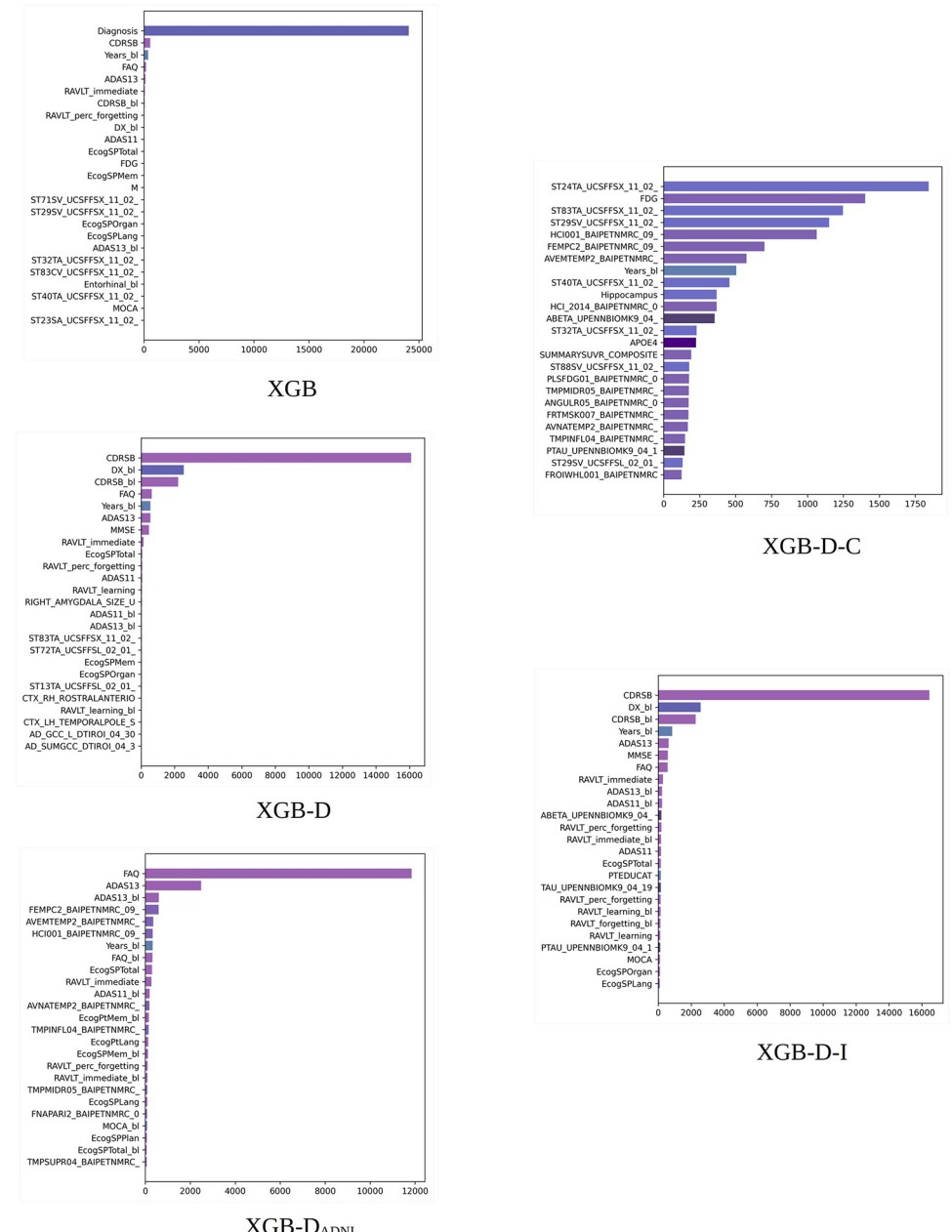

**Fig 1. XGB built-in importance.** Total gain scores obtained by different XGB models. Scores averaged over 10 random experiments for the 100% training set. Long feature names have been trimmed for better legibility. The bars are colored by feature type (diagnosis, clinical, cognitive, MRI, PET, DTI, APOE4, and proteomic biomarkers).

(-D-D$_{ADNI}$), diagnosis and cognitive features (-D-C), and diagnosis and image-based features (-D-I). The models were built with balanced training sets by selecting the 100% of the under-represented class (AD). In addition, S1-S3 Tables in S1 File compare the performance of the models built with different sample sizes of the under-represented class (100, 50, and 25%, respectively).

**5.1.2 TADPOLE rank preservation.** Comparing the performance metrics obtained with the TADPOLE results in Table 2 with those of our models in Table 3, we find that *Frog*,

**Table 2. Performance reported in the TADPOLE Challenge on D4.** Performance obtained with the benchmark methods in D2 and D4.

| | Evaluation set (D4) | | |
|---|---|---|---|
| | rank | mAUC | BCA |
| *Frog* | 1 | 0.931 | 0.849 |
| *ThreeDays* | 2 | 0.921 | 0.823 |
| *EMC-EB* | 3 | 0.907 | 0.805 |
| | Evaluation set (D4) | | |
| | rank | mAUC | BCA |
| SVM | 30 | 0.836 | 0.764 |
| Mixed Effects APOE | 35 | 0.822 | 0.749 |
| Last visit | 44–45 | 0.774 | 0.792 |
| | Test set (D2) | | Evaluation set (D4) | |
| | mAUC | BCA | mAUC | BCA |
| SVM, 1818 feat. | 0.877 | 0.765 | 0.828 | 0.744 |
| SVM, 6 feat. | 1.000 | 1.000 | 0.855 | 0.792 |
| Last visit | 1.000 | 1.000 | 0.774 | 0.792 |

Left: performance reported in the TADPOLE Challenge on the evaluation set (D4) for the methods considered in this work (results shown in [16], Table 2). Right, top: performance of the benchmark methods on the evaluation set (D4) reported in the TADPOLE Challenge (results shown in [16], Table 2). Right, bottom: performance of the benchmark methods on the test (D2) and evaluation (D4) sets obtained with our data.

**Table 3. Performance obtained with the methods considered in this work on the test (D2) and evaluation (D4) sets.**

| | Test set (D2) | | Evaluation set (D4) | |
|---|---|---|---|---|
| | mAUC | BCA | mAUC | BCA |
| XGB, 100% | 0.998 (± 0.000) | 0.982 (± 0.006) | 0.909 (± 0.004) | 0.804 (± 0.003) |
| XGB-D, 100% | 0.945 (± 0.001) | 0.863 (± 0.009) | 0.897 (± 0.004) | 0.788 (± 0.010) |
| XGB-D$_{ADNI}$, 100% | 0.920 (± 0.005) | 0.833 (± 0.005) | 0.881 (± 0.005) | 0.795 (± 0.008) |
| XGB-D-C, 100% | 0.791 (± 0.008) | 0.680 (± 0.013) | 0.732 (± 0.011) | 0.646 (± 0.016) |
| XGB-D-I, 100% | 0.942 (± 0.004) | 0.873 (± 0.003) | 0.909 (± 0.006) | 0.820 (± 0.009) |
| RF, 100% | 0.976 (± 0.003) | 0.908 (± 0.008) | 0.894 (± 0.004) | 0.804 (± 0.008) |
| RF-D, 100% | 0.933 (± 0.003) | 0.832 (± 0.008) | 0.880 (± 0.005) | 0.776 (± 0.012) |
| RF-D$_{ADNI}$, 100% | 0.904 (± 0.005) | 0.795 (± 0.010) | 0.864 (± 0.006) | 0.759 (± 0.008) |
| RF-D-C, 100% | 0.774 (± 0.006) | 0.671 (± 0.013) | 0.721 (± 0.010) | 0.625 (± 0.016) |
| RF-D-I, 100% | 0.945 (± 0.003) | 0.878 (± 0.011) | 0.900 (± 0.004) | 0.803 (± 0.008) |
| SVM, 100% | 0.925 (± 0.002) | 0.815 (± 0.006) | 0.845 (± 0.003) | 0.757 (± 0.003) |
| SVM-D, 100% | 0.909 (± 0.002) | 0.785 (± 0.007) | 0.839 (± 0.003) | 0.739 (± 0.005) |
| SVM-D$_{ADNI}$, 100% | 0.893 (± 0.002) | 0.784 (± 0.008) | 0.830 (± 0.003) | 0.732 (± 0.007) |
| SVM-D-C, 100% | 0.773 (± 0.006) | 0.656 (± 0.011) | 0.721 (± 0.006) | 0.619 (± 0.011) |
| SVM-D-I, 100% | 0.917 (± 0.002) | 0.816 (± 0.009) | 0.877 (± 0.002) | 0.777 (± 0.093) |

Models built with different feature sets: whole, without clinical diagnosis (-D), without clinical diagnosis and ADNI features for diagnosis (-D-D$_{ADNI}$), without clinical diagnosis and cognitive features (-D-C), and without clinical diagnosis and image features (-D-I). The training sets are balanced and use 100% of the under-represented class (AD). For each method configuration, the mean and standard deviation of the mAUC and BCA obtained over 10 experiments are shown.

*ThreeDays*, and *EMC-EB* outperform our methods (on D4, mAUC of 0.931 vs. 0.909, 0.921 vs. 0.894, and 0.907 vs. 0.845, respectively). Frog differs from our XGB system in terms of the training set, set of features, imputation, use of augmentation, and use of different temporal windows. *ThreeDays* differs from our RF system in terms of the training set, set of features, imputation, and use of forest ensembles in solving pairwise problems. *EMC-EB* differs from our SVM system in terms of the training set and set of features. Thus, the performance improvement achieved by the winners depends more on the data used, feature selection, and mentioned *ad hoc* methodological modifications than on the underlying machine learning method.

Nevertheless, the results in Table 3 for the models trained with all the features show that the ranks obtained in D2 and D4 by our systems are consistent with the TADPOLE Challenge podium. XGB obtains the best mAUC and BCA metric values, closely followed by RF, while the metrics obtained by SVM are considerably lower. Therefore, either XGB or RF seems to be the most competitive method for the problem of AD/MCI/CN identification. RF was also the best-performing method in the exhaustive study presented for a different application in [33].

**5.1.3 Feature set selection and performance.** In D2, the influence of feature set selection on the performance metrics is considerable. The best results are obtained for the feature set including the clinical diagnosis. The feature sets excluding the diagnosis (-D) degrade the performance of the methods considerably even though the clinical diagnosis is partially obtained from the $D_{ADNI}$ features. For example, the mAUC for XGB decreases from 0.998 to 0.945. For RF, the mAUC decreases from 0.976 to 0.933. For SVM, the mAUC decreases from 0.925 to 0.909. A further decrease in the feature set with the diagnosis and $D_{ADNI}$ features further degrades the performance.

The methods with feature sets excluding diagnosis and image features (-D-I, i.e., including cognitive features) perform consistently better than those with feature sets excluding diagnosis and cognitive features (-D-C, i.e., including image features). For RF and SVM, excluding diagnosis and image features leads to slight improvement in performance compared to excluding diagnosis. For XGB, both performances are nearly identical.

In D4, the overall best results are obtained for the feature set excluding clinical diagnosis and image features (-D-I). The feature set including all the features achieves the second position, and removing the diagnosis and subsequent $D_{ADNI}$ features has the effect of slightly degrading the performance. In this case, the loss in performance with the -D-C feature set is considerable, i.e., it is lower than the last visit benchmark performance.

**5.1.4 Robustness to sample size selection.** The robustness of the results with respect to the selection of the training set size was assessed through 10 different experiments for each method and feature set. For each experiment, the data were selected randomly, except for the under-represented class in the 100% experiment. Recall that S1-S3 Tables in S1 File list the mean and standard deviation of the mAUC and BCA obtained by the methods in D2 and D4.

In general, the metrics show a small standard deviation in both D2 and D4 with variations in the third decimal position in the majority of the cases. Therefore, all the models obtain consistent results for the same configuration of sample and feature selection. The models trained with 25% of the samples of the under-represented class obtain metrics slightly smaller but close to those of the models trained with the whole under-represented class. Therefore, the performance achieved by the methods can be considered robust to these changes in data sampling. Now, we proceed with the interpretability study for the 100% sample sets.

## 5.2 Interpretability study

**5.2.1 XGB built-in importance scores.** Fig 1 shows the 25 most important features according to the total gain importance of XGB averaged over 10 random experiments for the

100% training sets. Recall that the legend of the feature names can be found at https://github.com/swhustla/pycon2017-alzheimers-hack/blob/master/docs/data_dictionary.md. The diagnosis is scored as the most important feature. The following features show a considerable total gain reduction.

The most relevant features include the diagnosis, age of each patient at baseline (`Years_bl`), and several measurements from cognitive tests (`CDRSB`, `FAQ`, `ADAS13`, `RAVLT`, `CDRSB_bl`, `DX_bl`, `ADAS11`, and `ECog`). Then, a number of measurements from anatomical structures obtained from the FreeSurfer cross-sectional pipeline (`UCSFFSX`) arise, interleaved with the ADNIMERGE image and other cognitive measurements. The anatomical structures include the amygdala (ST17SV), hippocampus (ST29SV), inferior and middle temporal regions (ST32TA, ST40TA), entorhinal region (ST83CV), and cuneus (ST23SA).

When the diagnosis is removed from the XGB models, `CDRSB`, the diagnosis at baseline `DX_bl`, and `CDRSB_bl` occupy the first, second, and third positions, respectively. The previously mentioned cognitive features increase their relative importance. `CDRSB` and `MMSE`, used in ADNI for establishing diagnosis, occupy the top positions. Different `UCSFFSX`, `UCSFFSL`, `UCBERKELEY`, and `DTI` features can be now found in the last top 25 positions.

When removing the features used in ADNI for establishing diagnosis, `FAQ` and `ADAS13` occupy the top positions. In this case, the `UCSFFSX` features are mostly replaced by the `BAI-PET` features. These features are now highly interleaved with cognitive data. The `BAIPET` features include FDG measurements from the posterior cingulate (FEMPC), different AV45 measurements from the temporal region (AVEMTEMP, AVNATEMP, TMPINFL), and the hypometabolic index (HCI). We can easily find studies in the literature on the role of PET imaging of these regions in the characterization of AD [34, 35]. The hypometabolic index has been also identified as a biomarker useful for the characterization of the disease [36].

When removing all cognitive data, the `UCSFFSX` and `BAIPET` features mostly occupy the top 25 positions. Some features of relevance for the disease, such as FDG-PET, the hippocampus volume, or APOE4, can be seen in the top 25 set. The features from the posterior cingulate, precuneus, temporal region, and hypometabolic index still appear in similar positions. The importance is distributed over a wide number of features. From the `UPENNBIOMK9` biomarkers, `ABETA` and `PTAU` are in the top 25 set.

Removing image data makes cognitive features prevail in importance, and they occupy the top 25 positions. The most important features are the features used for diagnosis in ADNI. Then, the `ADAS` and `RAVLT` features follow in importance. Finally, the `ECog` features occupy the last of the top 25 positions. In addition, the three `UPENNBIOMK9` features (ABETA, TAU, PTAU) are now in the top 25 positions.

Fig 2 shows the 25 most important features according to the total cover importance of XGB averaged over 10 random experiments for the 100% training sets. In general, we can see discrepancies among the features selected according to the total gain and total cover although there is agreement in the features selected in the first positions.

**5.2.2 RF built-in importance scores.** Fig 3 shows the 25 most important features according to the built-in importance of RF averaged over 10 random experiments for the 100% training sets. The most striking difference from the XGB results is that the importance of RF is distributed over a greater number of features. The diagnosis is again the most important feature, mostly followed by cognitive features (`CDRSB`, `CDRSB_bl`, `FAQ`, `MMSE`, `ADAS`s, etc.) Only one FreeSurfer `UCSFFSX` feature from the entorhinal region (ST24TA) occupies one of the last top 25 positions.

Removing the diagnosis and the features used in ADNI for diagnosis results in cognitive features similar to XGB occupying the top positions. However, RF seems to be more dependent on the information given by cognitive data compared to XGB.

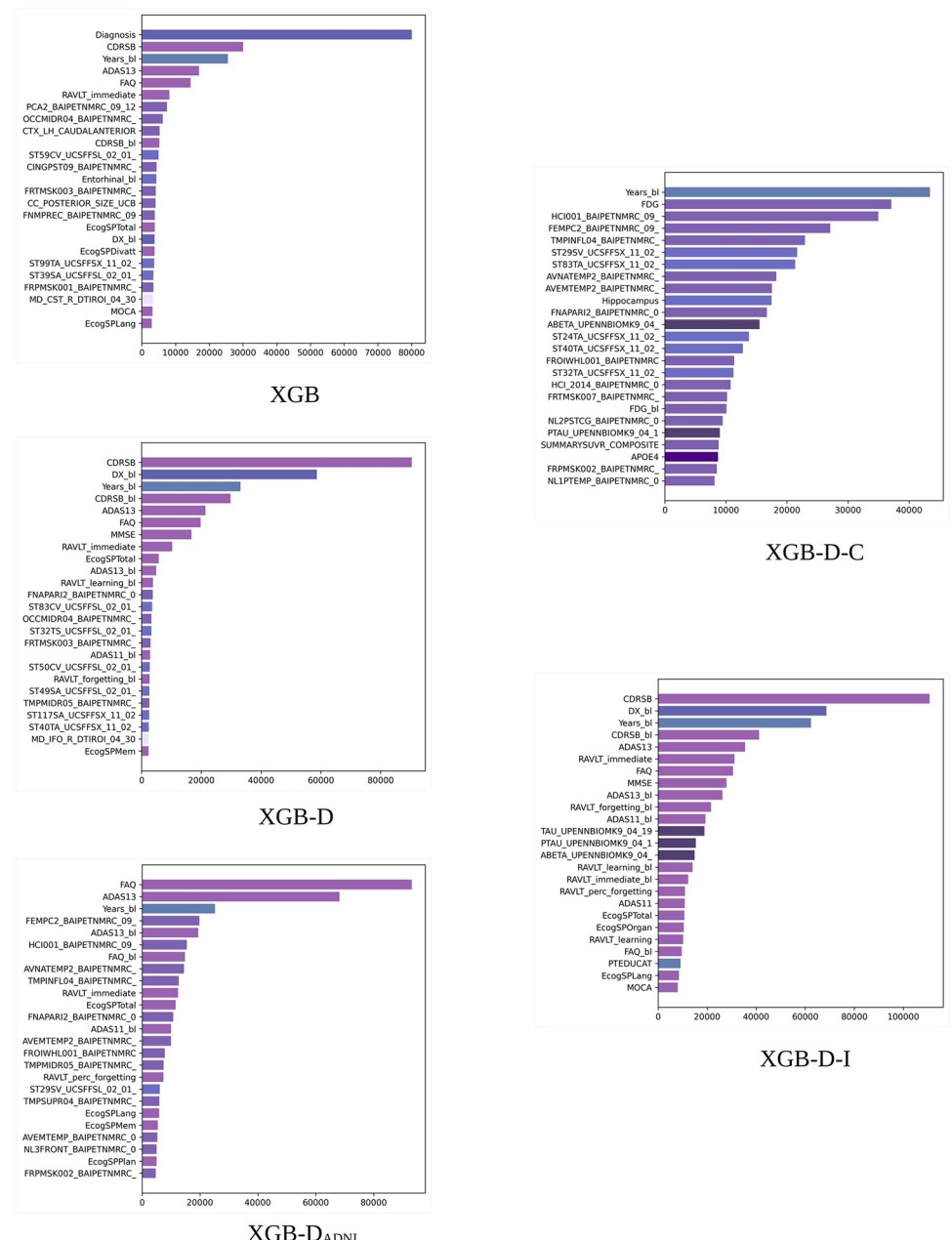

**Fig 2. XGB built-in importance.** Total cover scores obtained by different XGB models. Scores averaged over 10 random experiments for the 100% training set. Long feature names have been trimmed for better legibility. The bars are colored by feature type (diagnosis, clinical, cognitive, MRI, PET, DTI, APOE4, and proteomic biomarkers).

Upon removing all the cognitive data, the `UCSFFSX` features mostly occupy the top 25 positions. Some features of relevance for the disease, such as the hippocampus (ST29SV, Hippocampus, ST88SV), FDG-PET (FDG), entorhinal volume (ST24TA, ST83TA, Entorhinal), and middle and inferior temporal regions (ST40TA, ST32TA), can be seen among these features. Removing image features results in a configuration of top cognitive features nearly identical to XGB. Here, the `UPENNBIOMK9` features do not appear in the top 25 positions.

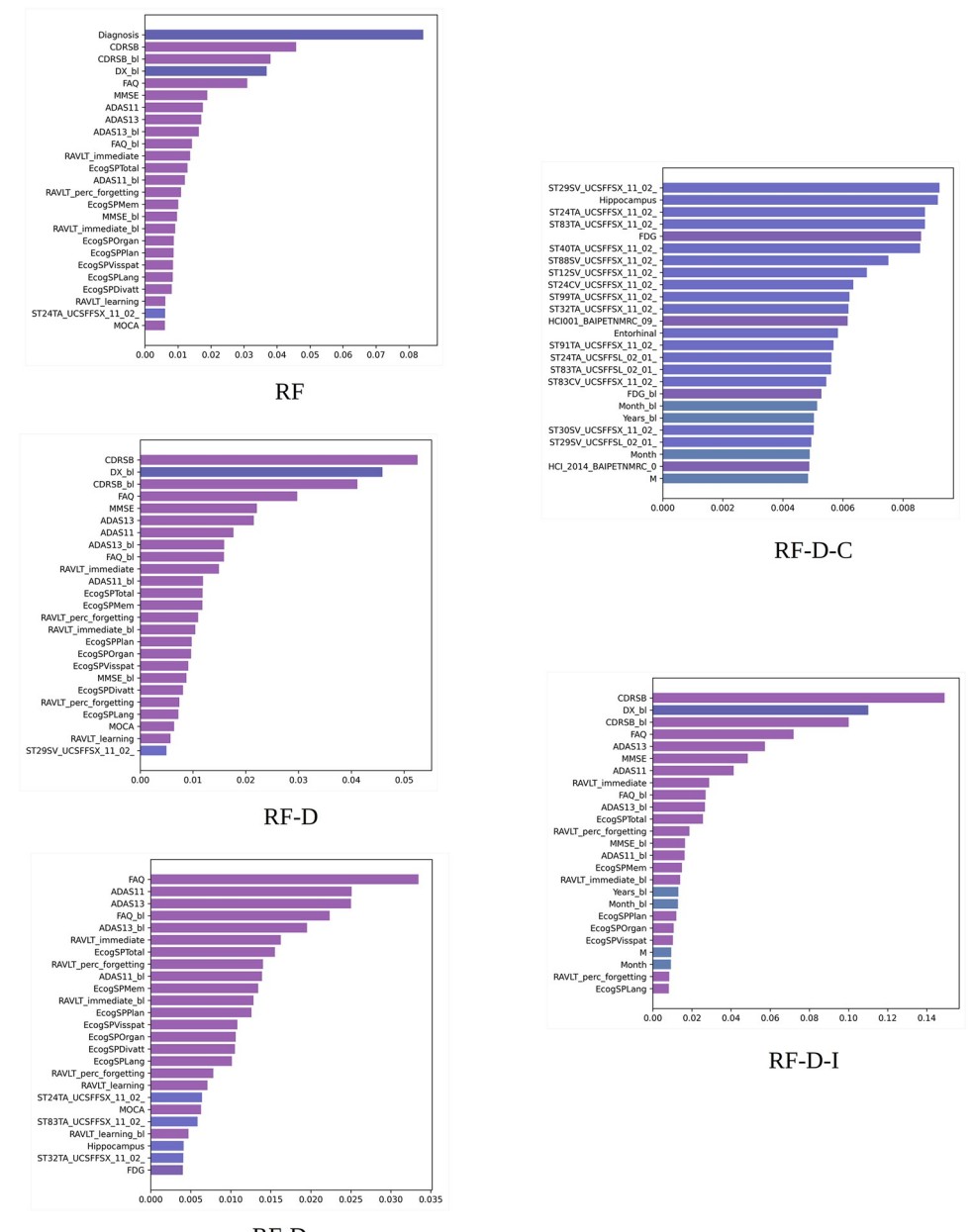

**Fig 3. RF built-in importance.** Importance scores obtained by different RF models. Scores averaged over 10 random experiments for the 100% training set. Long feature names have been trimmed for better legibility. The bars are colored by feature type (diagnosis, clinical, cognitive, MRI, PET, DTI, APOE4, and proteomic biomarkers).

**5.2.3 SHAP-based explainability using bar plots.** Figs 4–6 show the 25 most important features according to the mean absolute SHAP values in one of the experiments for the 100% training sets and the models including the whole feature set. The diagnosis is recognized by SHAP as the most important feature. The relative importance compared with subsequent features is remarkably greater for XGB and SVM than for RF.

For XGB, the most relevant features include the diagnosis, age of each patient at baseline (`Years_bl`), some features characterizing the cognitive condition (`CDRSB`, `ADAS`, `FAQ`,

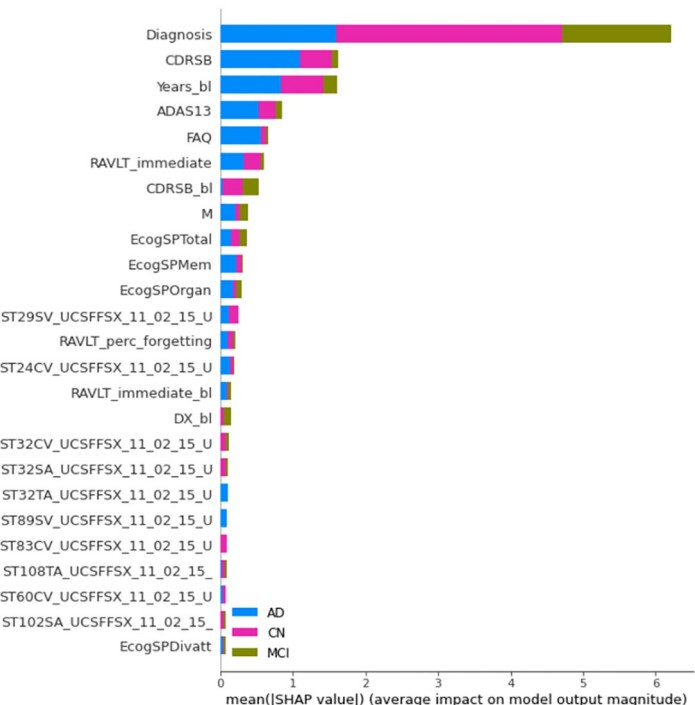

**Fig 4. Mean absolute SHAP values computed from the XGB systems.** Models trained with all features. Note that the label coloring for the CN, MCI, and AD classes is not consistent among experiments owing to rigid SHAP implementation. Long feature names have been trimmed for better legibility.

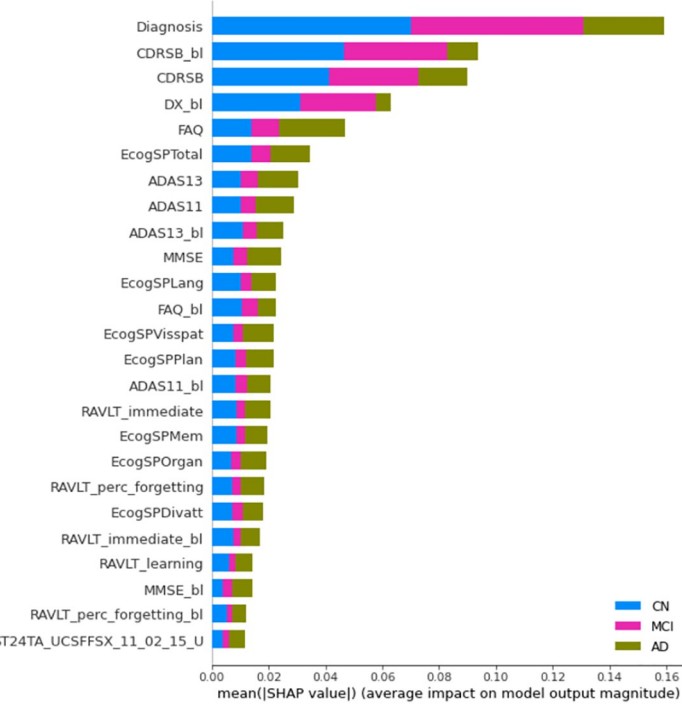

**Fig 5. Mean absolute SHAP values computed from the RF system.** Same legend than Fig 4.

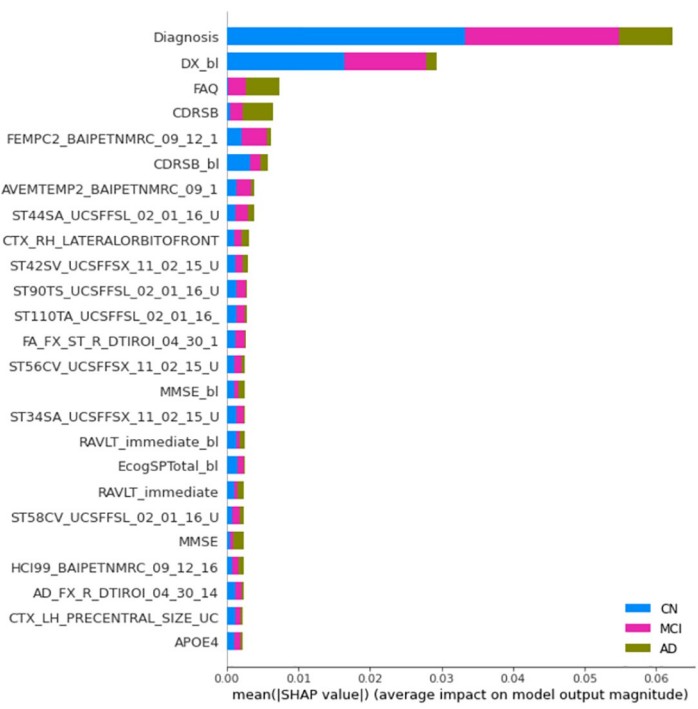

**Fig 6. Mean absolute SHAP values computed from the SVM systems.** Same legend than Fig 4.

`RAVLT`, `CDRSB_bl`, and `ECog`), and measurements from FreeSurfer cross-sectional segmentations (`UCSFFSX`) . These measurements refer to the volume, surface area, or cortical thickness of regions related to AD, such as the hippocampus (ST29SV), entorhinal region (ST24CV, ST83CV), inferior temporal region (ST32S, ST32T), and temporal pole (ST60).

SHAP feature ranking in the first positions is more coherent with the XGB total gain compared to the total cover built-in importance results. In addition, SHAP provides information regarding the importance of each feature to the different classes. For the identification of CN individuals, the diagnosis is mainly the most relevant feature. For the diagnosis of AD individuals, the diagnosis, `CDRSB`, `Years_bl`, `ADAS13`, and `FAQ` are the most relevant features. Thus, the relevance of `CDRSB_bl` and `DX_bl` in the diagnosis of AD is quite low. As these features are not useful for the correct classification of converters, it seems that the system can identify the diagnosis and `CDRSB` as preferable features for AD diagnosis. For the identification of MCI individuals, the importance of diagnosis is much greater than the importance of other features.

For RF, the most relevant features are all cognitive, including `CDRSB_bl`, `CDRSB`, `DX_bl`, `FAQ`, `ECog`, `ADASs`, `MMSE`, and `RAVLT`. The only image-based feature is the cortical thickness average of the left entorhinal region (ST24T). As with XGB, the diagnosis is more relevant for the identification of CN and MCI individuals. In contrast to the XGB results, the importance of the features is much more distributed among the top 25 features for the CN and AD classes. The built-in importance results showed a similar set of features in similar positions.

For SVM, the diagnosis and `DX_bl` are the most relevant features, followed by `FAQ`, `CDRSB`, and `CDRSB_bl`. Subsequently, a number of image-based features are found, including the `BAIPET`, `FFSL`, `AV45`, `FFSX`, and `DTI` features. These features refer to regions such as the posterior cingulate gyrus (FEMPC2), temporal area (AVETEMP2), parahippocampal

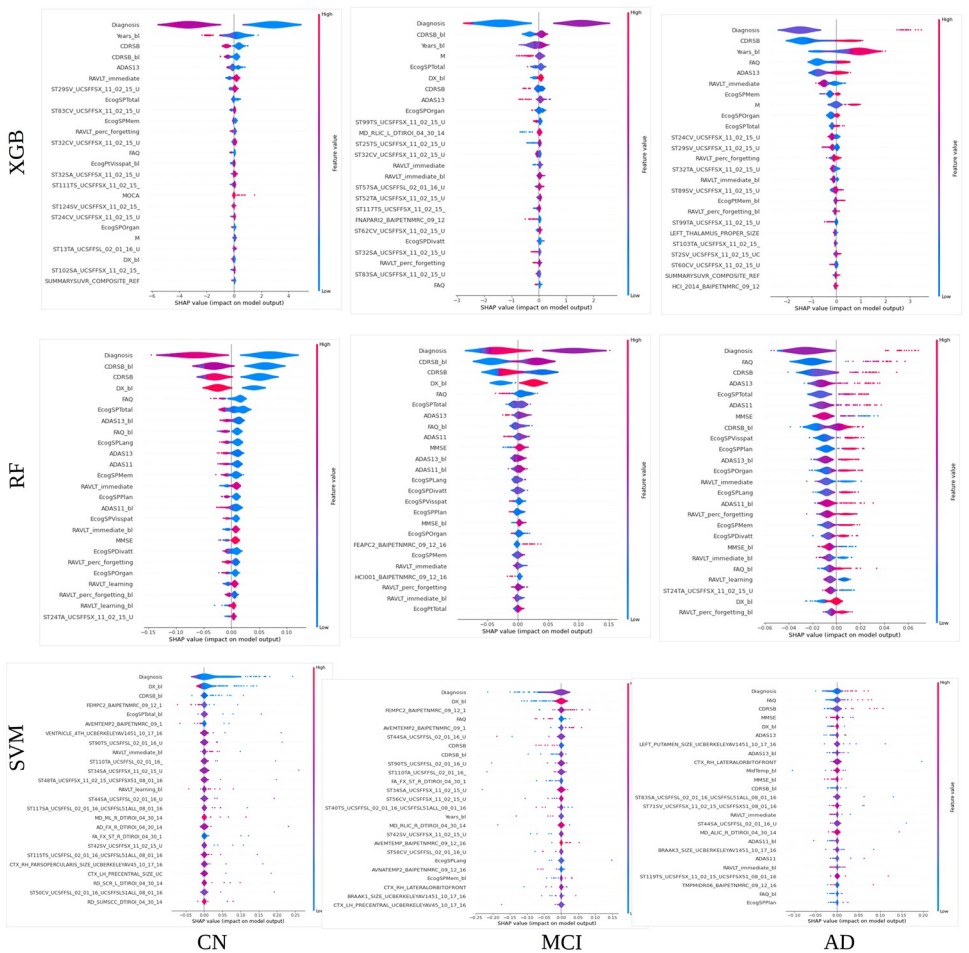

**Fig 7. XGB, RF, and SVM explainability.** The violin plots show the SHAP value impact on the estimation of the probabilities for each class of the XGB, RF, and SVM models. Models trained with all the features. Long feature names have been trimmed for better legibility.

region (ST44SA), pallidum (ST24SV), parietal and precentral areas (ST90TS, ST110TA), fornix (FAFXSTR), superior frontal area (ST56CV), isthmus (ST34SA), and superior temporal area (ST58CV). Some more cognitive features such as `MMSE` and `RAVLT_immediate` can be found in baseline-current visit pairs. APOE4 is the last feature in the top 25 positions. Thus, the AD class is mostly explained by the diagnosis, `FAQ`, and `CDRSB`, and, to a lesser extent, by `MMSE`.

**5.2.4 SHAP-based explainability using violin plots.** Fig 7 shows the impact of each feature on the different model outputs for each class. The impact is represented by violin plots, as is customary in SHAP analysis. Comparing the distribution of the violin plots of RF and XGB, we can see that the contribution of the first features to the probabilities is polarized in two modes for a greater number of features. Thus, for RF, the values of all these features are more strongly considered in the classification. For SVM, the distribution of the violin plots shows a single mode for all features.

The impact representation of the diagnosis feature suggests that low diagnosis values favor high probabilities for the CN class ($p(CN)$), and low probabilities for the MCI and AD class ($p(MCI)$ and $p(AD)$). High diagnosis values favor low $p(CN)$ and high $p(AD)$. Mean diagnosis

values favor high $p(MCI)$. As the labels assigned to each class are 0 for CN, 1 for MCI, and 2 for AD, all the models can capture the relationship between labels and future status.

The violin plots of `Years_bl` for XGB indicate that the system considers the information of the age when the patient is included in the study. The greater the age, the greater is the $p$ ($AD$). This is consistent with the knowledge that AD is linked with age. The model also considers the months from baseline (`M`) as important.

Regarding the `CDRSB_bl` and `CDRSB` features, the lower the values, the greater is $p(CN)$ and viceversa for $p(AD)$. High values of $p(MCI)$ are favored by mean values of `CDRSB`. The scores reflect the Hughes Clinical Dementia Rating (CDR), where a CDR of zero indicates that the subject is healthy and gradually increases until values of 3, indicating that the subject suffers from a serious cognitive decline.

Regarding cognitive scores, a low value in `FAQ`, `ADAS13`, and `ADAS11` and a high value in `RAVLT`s favor high $p(CN)$, and viceversa for AD. It has been reported that higher Adas-Cog and lower RAVLT scores suggest greater impairment [37]. Regarding MRI features, medium to high volume values favor high values of $p(CN)$. This indicates that the systems can associate atrophy with high values of $p(AD)$ and vice versa for $p(CN)$.

In summary, the violin representation of the SHAP values shows that the three models can establish a relationship between feature values and diagnosis probability consistent with clinical knowledge for the top 25 features. The distribution of the SHAP value impact tends to be bi-modal for the most important features of the best-performing methods.

**5.2.5 Removing diagnosis, cognitive, and image features.**   Fig 8 shows the reassignment of importance to the remaining features of the XGB, RF, and SVM models after progressively removing the diagnosis (-D), the features used by ADNI in diagnosis (-D-D$_{ADNI}$), cognitive (-D-C), and image (-D-I), features. The different systems rebalance the importance given to the features, and the contribution becomes less biased to a single prevalent feature.

After removing diagnosis, the features used in ADNI for the diagnosis occupy top positions, as with the built-in importance study. `CDRSB`s features are found to be much more relevant than `MMSE`. For XGB, image-based features from the hippocampus and entorhinal region prevail. For SVM, most image-based features prevail.

After removing the diagnosis and D$_{ADNI}$ features for XGB and RF, `FAQ`s the features become the most important, followed by cognitive features such as `ADAS`, `RAVLT`, and `ECog`. For XGB, `Years_bl` still occupies the top positions. For SVM, the behavior is slightly different from XGB and RF. `FAQ` still tends to occupy top positions. However, image-based features such as `BAIPET`, `FFSL`, and `FFSX` increase their relevance to the model. These features relate to regions such as the posterior cingulate gyrus for FDG-PET and the temporal region for AV45-PET. The unknown region (ST123TA) also appears as relevant for SVM.

After removing all the cognitive features, the system again rebalances the importance given to the features, and image-based features occupy higher positions. Some MRI and PET measurements such as `FDG`, the hippocampus volume, `FDG_bl`, and the entorhinal volume now increases their importance. Both XGB and RF agreed on the importance of the anatomical features related to the entorhinal region, hippocampus, middle temporal region, and amygdala. In addition, both methods agreed on the importance of the hypometabolic index, FDG measurements of the posterior cingulate gyrus, and AV45 measurements of the temporal region. For SVM, the FDG-PET measurements of the posterior cingulate gyrus and the AV45-PET measurements of the temporal region were the most important features. `APOE4` reaches the top 5 positions in importance. After removing image-based features, cognitive features mostly occupy the top 25 features. For XGB, the `TAU` and `ABETA` biomarkers are in the top 25 positions. For SVM, the `ABETA` biomarker is also shown in the last top 25 positions.

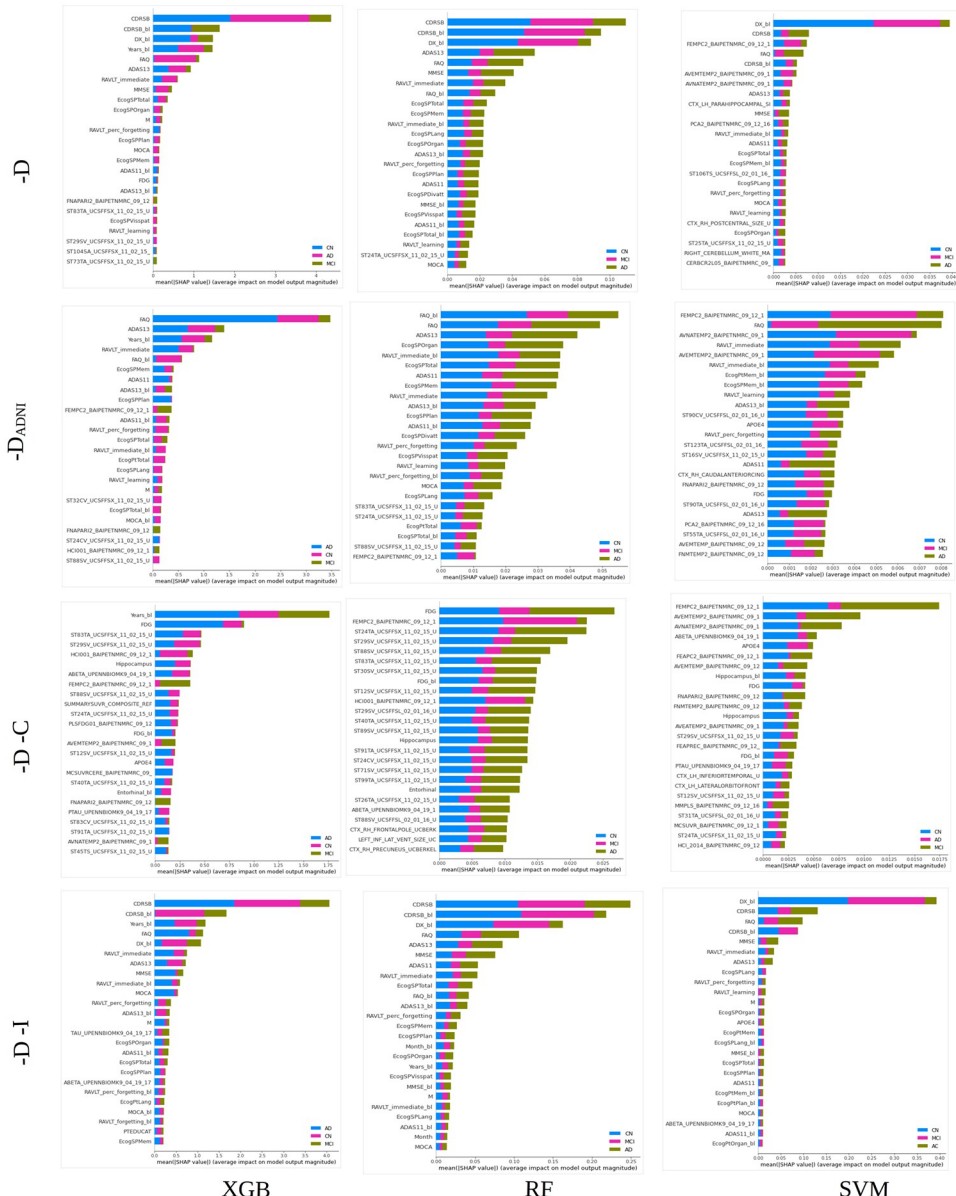

**Fig 8. Mean absolute SHAP values computed from the XGB, RF, and SVM systems with feature ablation.** First and second rows: models trained after removing diagnosis (-D) and features used in ADNI for diagnosis (-D-D$_{ADNI}$). Third and fourth rows: models trained after removing cognitive (-D-C) and image-based (-D-I) features. Note that the label coloring is not consistent among experiments owing to rigid SHAP implementation. Long feature names have been trimmed for better legibility.

The supplementary material shows the violin plots of the impact of the features on the model outputs for the different feature sets (S2-S4 Figs in S1 File).

## 5.3 Robustness to sample selection

Figs 9–11 show the frequency of the top 25 features over 10 experiments for the 100% training set and the different feature sets. The results can be found in the csv files of the supplementary material: S1–S3 Data, respectively. For XGB and the whole feature set, around 50% of the top

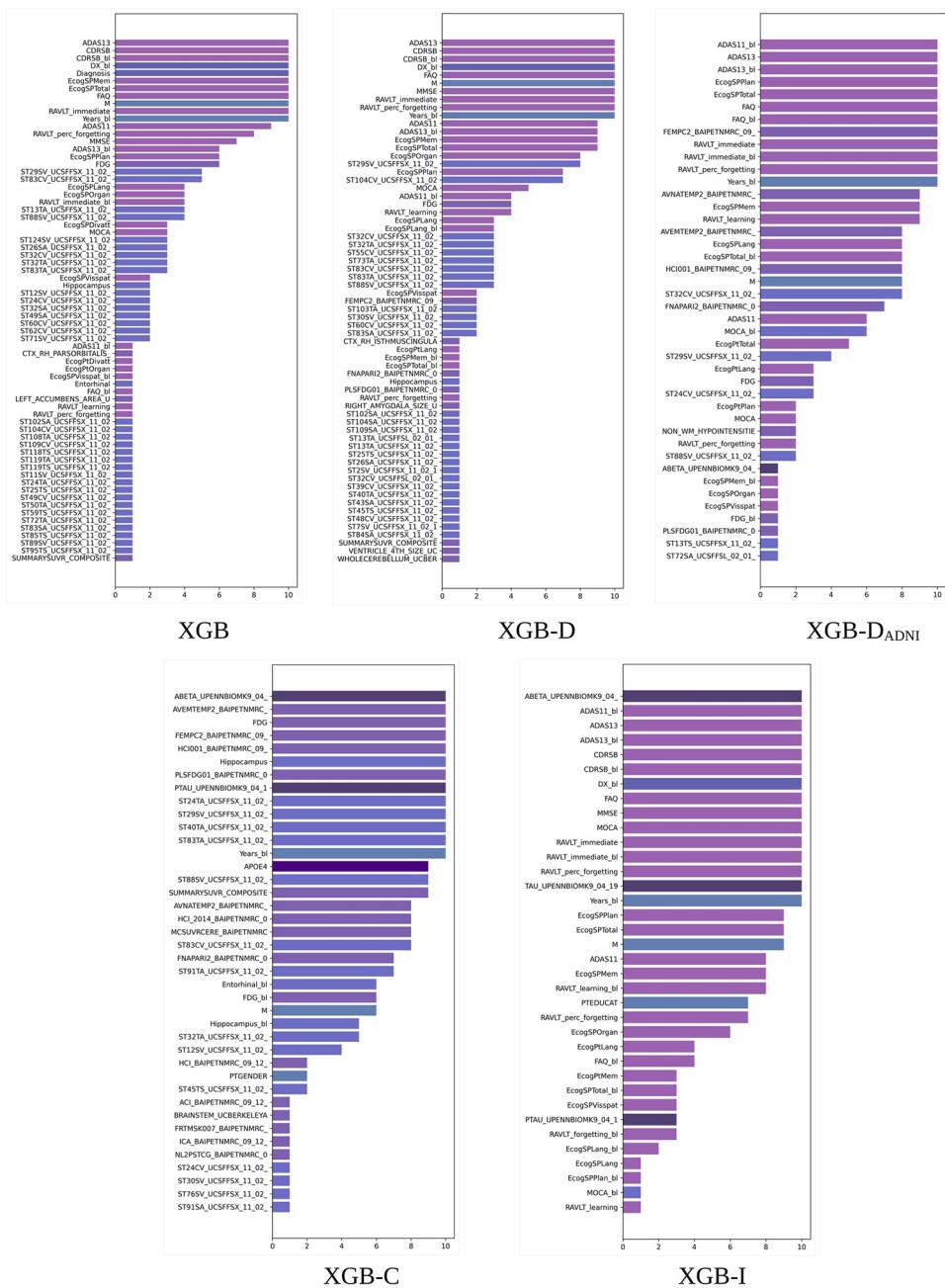

**Fig 9. Robustness of SHAP importance for XGB.** Frequency of the top 25 features over 10 random experiments for the 100% training set. The bars are colored by feature type (diagnosis, clinical, cognitive, MRI, PET, DTI, APOE4, and proteomic biomarkers).

25 features are consistently shown in the top 25 positions. These features include the diagnosis, `Years_bl`, and cognitive features. The most frequent `UCSFFSX` features are volume and area measurements from the hippocampus, entorhinal region, superior temporal sulcus, and inferior temporal region. Upon subsequently removing the diagnosis, ADNI, cognitive, and image-based features, the number of features consistently increases. For the -D and -D-D$_{ADNI}$ sets, the most consistent features are mostly cognitive. For the -D-C feature set, apart from

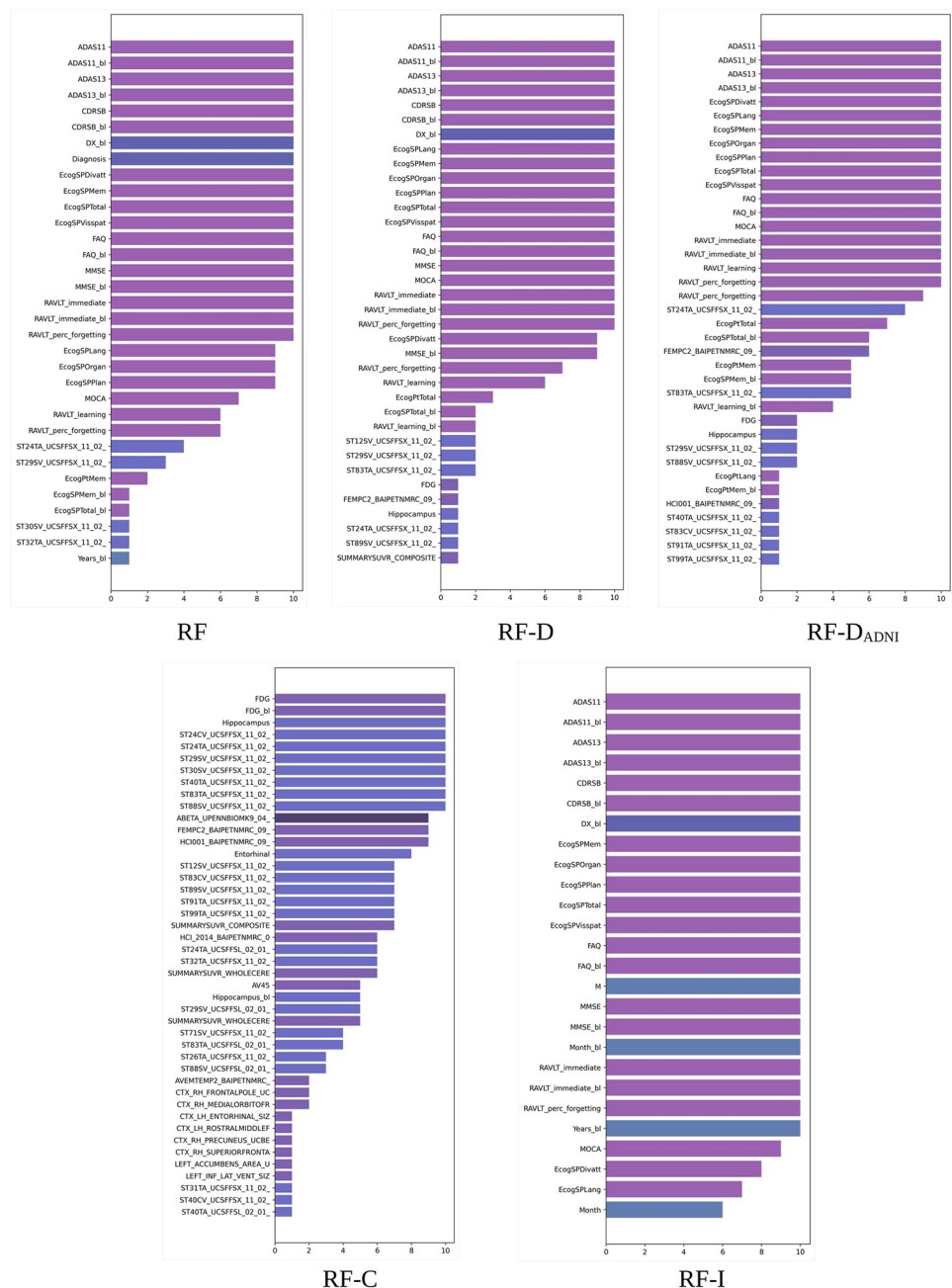

**Fig 10. Robustness of SHAP importance for RF.** Frequency of the top 25 features over 10 random experiments for the 100% training set. The bars are colored by feature type (diagnosis, clinical, cognitive, MRI, PET, DTI, APOE4, and proteomic biomarkers).

measurements related to the entorhinal region, hippocampus, and middle temporal region, the most frequent features in the top 25 positions are FDG, PET AV45 measurements in the temporal region, the hypometabolic convergence index, and `APOE4`, `ABETA`, and `PTAU` values. For the -D-I feature set, the top 25 most selected features are `CDRSB`, `FAQ`, `RAVLT`, `ADAS` scorings, etc., and not cognitive features such as `TAU`, `ABETA`, and `Years_bl`.

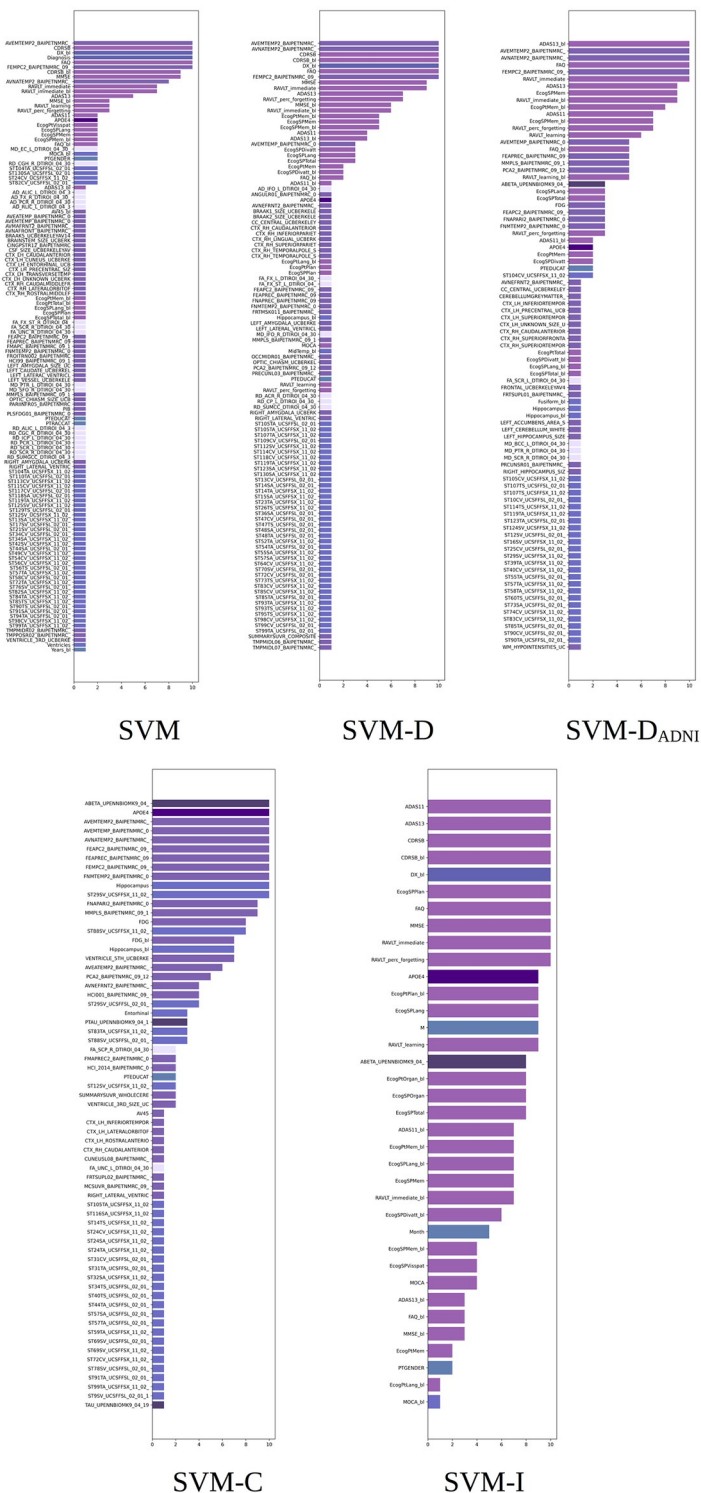

**Fig 11. Robustness of SHAP importance for SVM.** Frequency of the top 25 features over 10 random experiments for the 100% training set. The bars are colored by feature type (diagnosis, clinical, cognitive, MRI, PET, DTI, APOE4, and proteomic biomarkers).

For RF, the number of features selected in the top 25 positions is more robust and focused on cognitive features. For the -D-C feature set, the measurements from the entorhinal region, hippocampus, inferior lateral ventricle, and middle temporal region can be consistently seen in the top 25 positions. For the -D-I feature set, the consistency reaches nearly 100% of the features in the top 25 positions with mostly cognitive features.

For SVM, only 25% of the top 25 features are consistently shown in the top 25 positions. These features include the diagnosis, diagnosis at baseline, BAIPET, and cognitive features. For the whole, -D, and -D$_{ADNI}$ feature sets, feature importance is quite inconsistent. For the -D-C and -D-I feature sets, feature importance is much more robust. For the -D-C feature set, the BAIPET features are mostly selected together with APOE4 and ABETA UPENNBIOMK9. For the -D-I feature set, the most relevant cognitive features throughout our interpretability study are mostly selected in the top 25 feature set.

# 6 Discussion of our interpretability study in the context of the state of the art

## 6.1 TADPOLE Challenge feature importance

According to the global study of feature importance performed in TADPOLE, ABETA from the UPENNBIOMK9 features, image features for DTI, and APOE status contributed toward the best accuracy. The results obtained in our interpretability study did not corroborate these findings for the top three methods. ABETA biomarker only appeared among the best important features in the XGB-D-C, XGB-D-I, RF-C, and SVM-D-I models. We observed that the UPENNBIOMK9 features are considered as relevant for the models obtained with the -D-C or the -D-I sets. Therefore, the UPENNBIOMK9 features may be of importance because many of the TADPOLE Challenge methods used partial sets of features. In addition, DTI features hardly appeared in the built-in feature importance plots. APOE4 was only relevant when XGB was deprived of from cognitive information or SVM was deprived of from ADNI diagnosis, cognitive, or image-based features. The authors of *Frog* reported that the XGB built-in importance scores suggested that the MRI features play a greater role in models trained with long forecast windows. This finding was also obtained from our XGB model, where 10 out of 25 features belonged to the UCSFFSX family.

Meanwhile, the TADPOLE Challenge results indicated that image features from tau PET (AV1451), amyloid PET (AV45), and FDG-PET (FDG) contributed toward the worst accuracy. However, our results showed that the FDG feature was the most important feature in the RF-D-C model and among the most important features in XGB-D-C and SVM-D-C. In addition, FDG_bl was also present in the top 25 feature set. AV45 appeared frequently in the top 25 set of the RF-C model. Some FDG and AV45 PET features were relevant in models with the -D-C feature set.

The feature selection study performed in [17], the closest journal version of *ThreeDays*, identified the diagnosis, RID (patient identifier), AGE, MIDTEM (middle temporal gyrus), CDRSB, GENDER, FAQ, APOE, and MMSE as the most relevant features for accurate identification of the AD/MCI/CN subjects with RF. Accordingly, our study has corroborated the importance of the diagnosis, CDRSB, FAQ, and MMSE.

The ablation study in [23], the journal version of *CBIL*, highlighted the importance of the diagnosis and CDRSB features in high-performance systems. The authors observed significant drops in the evaluation metrics upon removing these features from their recurrent neural network (RNN) model. Our study also involved this loss in accuracy for the models with the -D and -D-* feature sets.

## 6.2 Feature importance outside the TADPOLE Challenge

**6.2.1 SHAP in RF-based multilayer multimodal detection and prediction model.** Outside the TADPOLE Challenge framework, El–Sappagh et al. [24] recently proposed the use of SHAP values in a RF-based multilayer multimodal detection and prediction model. The detection layer performs the classical classification into CN, MCI, and AD. The authors reduced the feature set to a total of 28 baseline features and the results were obtained with a single model optimized with classical cross-validation. Their feature set did not include the diagnosis, as it is undesirable to include the label as a feature for the classification problem. In addition, some features not present in the TADPOLE Challenge were used in the generation of the model.

The study identified CDRSB as the most influential feature for CN and MCI, followed by MMSE for AD. From the TADPOLE Challenge features used in that study, the model recognized cognitive and UPENNBIOMK9 features as the most relevant features. SHAP violin plots for AD showed MOCA, FAQ, ADAS13, and ADAS11 in the top relevant positions. Subsequently, ABETA, PTAU, and TAU were also shown as influential. RAVLT_immediate was also present in the relevance plot. Reasoning regarding feature values and clinical knowledge showed that the model correctly favored the probability of the different classes. For example, high values of CDRSB favored $p(AD)$ whereas low values of CDRSB and high values of MOCA favored $p(CN)$.

In our RF models, cognitive features prevailed in the top positions. In the -D-I model, whose feature configuration is probably the closest to that of El-Sappagh's model, CDRSB was also the most influential feature and FAQ, ADAS13, MMSE, and ADAS11 were shown in the top positions. The violin plots showed similar behavior in favoring the probabilities of the different classes. However, MOCA was found at the 25th position and UPENNBIOMK9s were found beyond the top 25 set. As seen in our ablation study, the importance given by the models to the different features strongly depends on the feature set used to generate the model. It may be possible that the features not present in the TADPOLE Challenge combined with feature selection are responsible for the rise of MOCA and UPENNBIOMK9 as relevant features for El–Sappagh's model.

**6.2.2 Systematic, quantitative, and critical review of machine learning in predicting the progression of MCI.** The exhaustive review in [38] showed that including cognitive and FDG features significantly improved the predictive performance of the methods compared to not including them. Other modalities, especially MRI-based features, did not show a significant effect. The authors argued that the good performance of cognitive assessments cast doubts on the wide use of imaging for predicting the progression of AD and suggested exploring further fine domain-specific cognitive evaluations.

Our study corroborated the importance given by the different models to the cognitive features. In fact, RF mostly relied on cognitive information. In addition, FDG was identified as a relevant feature when cognitive information is substantially missing. Although the use of cognitive features seems to be critical for an accurate and trustable decision support system in clinical practice, it should be noted that cognitive features are useful only once the patients have started to show some cognitive decline. According to the results of our study, we believe that different sets of features can be used depending on the stage of the disease for accurate computer-aided diagnosis systems. For early diagnosis, we believe that one should discard cognitive information and further study MRI, PET, DTI, biomarkers, and genetic features. Our study has shown that XGB and RF can identify the relevant information according to clinical knowledge for different sets of features while SVM is less accurate and seems to be less reliable. We believe that SHAP values may provide suitable criteria for feature selection.

## 6.3 Diagnosis as a feature

The clinical diagnosis has been identified in our study as the most relevant feature for all the models. The difference in importance with the following features is considerable for all the methods. In [17] and [23], the relevance of the diagnosis to the accuracy of the prediction for their random forests and RNN systems has been identified.

Table 4 lists the top 25 TADPOLE Challenge methods and indicates whether they used diagnosis in the feature set. We can see that the best-performing methods consistently used the diagnosis. The first two methods that did not use diagnosis were *EMC1-Std* (disease progression model combined with SVM, mAUC = 0.898) and *CBIL* (RNN, mAUC = 0.897). These methods ranked eighth and ninth, respectively. Although the diagnosis was explicitly included in the feature set of the majority of the best-performing TADPOLE methods, we believe that the inclusion of such a feature in a prognostic model may be questionable; moreover, the diagnosis values from the previous visits provide the values of the predicted labels in the training set. Methods not including the diagnosis in their feature set and achieving an accuracy close to 0.90 should be reconsidered as potential well-performing methods for the AD/MCI/CN identification problem.

## 7 Discussion and conclusions

We investigated the performance and interpretability of the three best-performing methods in the TADPOLE Challenge for the prognosis of clinical AD status. We used the same data

**Table 4. TADPOLE Challenge top 25 methods and use of diagnosis.**

| Method | Rank | Use diagnosis |
|---|---|---|
| Frog | 1 | Yes |
| ThreeDays | 2 | Yes |
| EMC-EB | 3 | Yes |
| Apocalypse | 7 | Yes |
| GlassFrog-Average | 5 | Yes |
| GlassFrog-SM | 5 | Yes |
| GlassFrog-LCMEM-HDR | 5 | Yes |
| EMC1-Std | 8 | No |
| CBIL | 9 | No |
| CN2L-RandomForest | 10 | Yes |
| EMC1-Custom | 11 | No |
| BGU-LSTM | 12 | No |
| DIKU-GeneralisedLog-Custom | 13 | No |
| DIKU-GeneralisedLog-Std | 14 | No |
| ARAMIS-Pascal | 15 | No |
| VikingAI-Sigmoid | 16 | Yes |
| Tohka-Ciszek-RandomForestLin | 17 | Yes |
| IBM-OZ-Res | 18 | Yes |
| BORREGOTECMTY | 19 | Yes |
| VikingAI-Logistic | 20 | Yes |
| lmaUCL-Std | 21 | Yes |
| lmaUCL-Covariates | 22 | Yes |
| Chen-MCW-Stratify | 23 | No |
| AlgosForGood | 24 | No |
| lmaUCL-halfD1 | 26 | Yes |

processing choices for fair comparison of the three methods. Interpretability was studied by combining feature ablation with feature importance assessment. We investigated the built-in feature importance metrics, when available, and we computed SHAP values from the models. SHAP is a recent tool for XAI that allows robust study of per-class feature significance and establishes the impact of each feature in the output probabilities. It has been shown to be a useful tool for answering key questions in terms of increasing or decreasing our confidence in conventional machine learning methods that occupy the top positions in the TADPOLE Challenge.

Our models preserved the top three ranks achieved in the TADPOLE Challenge, i.e., gradient booster ranked in the first position, random forest in the second one, and support vector machines closed the top three. However, the performance achieved on the evaluation set D4 was consistently lower than that reported by the TADPOLE Challenge results. We believe that these differences are due to the data used, feature selection, and methodological modifications to the standard machine learning method. For example, we believe that the accuracy of *Frog* is due to the combination of gradient booster with feature selection and data augmentation. *ThreeDays* is a model ensemble of two models specialized in binary classification problems with aggressive selection of features that may be the key to improved accuracy.

The accuracy of the models was robust to the selection of the training data. The inclusion of 1818 features did not degrade the performance of the models. It seems that once the diagnosis is removed, a number of cognitive features are required for acceptable performance. Otherwise, the performance of the system is considerably degraded. This is consistent with results shown by the state of the art. For example, the CadDementia Challenge winner [13] obtained a limited AUC of 0.788 in the problem of AD/MCI/CN classification from MRI image-based features. Spasov et al. [39] obtained an AUC of 0.925 by combining cognitive and structural MRI images for the problem of sMCI/pMCI identification. The AUC decreased to 0.880 for cognitive features and to 0.790 for MRI images.

From the study of feature importance, we found that the diagnosis is the most significant feature for the three methods. In general, the three methods give importance to features that have been shown to be related to the disease in the state of the art. Apart from the diagnosis, cognitive features tend to occupy the top positions in different feature sets. From the features used in ADNI for establishing the diagnosis, CDRSB is shown to be the most relevant cognitive feature. Focusing on image-based features, the top positions are occupied by measurements from the hippocampus, entorhinal region, and the middle temporal region. Therefore, the methods can identify anatomical structures affected by the disease among more than 700 MRI-based features. Similarly, for PET-based features, FDG-PET measurements in the posterior cingulate gyrus, AV45-PET measurements in the temporal region, or the hypometabolic index are identified as relevant among more than 300 features. These image-based biomarkers have been previously established as relevant in different studies [40–42].

In view of the variability of the XGB-selected features in the top 25 set, we recommend using XGB with model ensembles. Thus, the resulting system would consider the information of the most important features for each ensemble and make better-informed decisions. Meanwhile, our study reported that RF models mainly rely on cognitive information. As it is widely recognized that AD is a multimodal disease, clinically reliable machine learning methods should not be supported only by features from a single domain. The results in this study may indicate that a trustable RF system from the clinical point of view may be modeled with a reduced set of cognitive features to allow other features to increase their importance in the decisions. For SVM, our study reported that the importance given to different features strongly changes among different bootstraps. This result makes us question the suitability of SVM for our computer-aided diagnosis problem.

At this point, our study has provided sufficient insights to answer our initial questions:

- Why did the best methods achieve the best accuracy metrics? The best methods consistently used the diagnosis in the feature set. Some of the preprocessing decisions and methodological modifications boosted the mAUC beyond 0.90. Not including the diagnosis in the feature set limits the value of mAUC to 0.89 for the best-performing method.

- How can we quantify the contribution of each feature toward achieving the best accuracies? The built-in importance of tree-based methods and SHAP was shown to be useful for quantifying the contribution of the features toward the accuracies. SHAP provides different ways of representing information that further elucidate feature importance.

- Which features were the most meaningful for the best methods? The diagnosis was the most meaningful feature, with differences among the three methods. Different subsequent features arise depending on the method. For XGB, these features are cognitive features mixed with MRI image-based features. For RF, these features are mostly cognitive. For SVM, cognitive features are highly interleaved with a variety of image-based features.

- Do the best methods use information coherent with clinical knowledge? The violin plot representation of SHAP provided a way to verify that the values of the features are used consistently with clinical knowledge by different systems. Even if the model is built from a large image-based feature set, the methods have been shown to be able to select the most relevant features for making informed decisions.

One shortcoming of our study, inherited from the original TADPOLE Challenge setup, is that the test and evaluation sets originate from the same data distribution used for training the models. Thus, the evaluation and interpretability results of our models may not be preserved with different test and evaluation datasets. Therefore, our findings should be corroborated using different external clinical datasets.

In summary, our study showed the ability of the top two TADPOLE Challenge methods to correctly identify the features most relevant to the prognosis of disease status, thereby boosting our confidence in the potential of these systems for computer-aided prognosis. In the future, we will further explore SHAP features for the improvement of the explanations toward improved interpretability, causability, and actionable explainability. Furthermore, we will extend the study to other TADPOLE Challenge methods with the potential to better adapt to the temporal nature of the prognosis problem that scored lower than the best-performing methods (e.g., *CBIL*) with the aim of establishing a fair ranking within a consistent framework. In addition, we will replace the image-based features with the original images to study the interpretability of the resulting systems in order to build systems capable of extracting useful high-level features invisible to the human eye.

## Supporting information

**S1 File. Accompanying document with some results complementing the study.** Referred in the manuscript as supplementary material.
(PDF)

**S1 Data. Frequency of the top 25 features over 10 experiments for the 100% training set and the different feature sets.** File obtained with XGB.
(CSV)

**S2 Data. Frequency of the top 25 features over 10 experiments for the 100% training set and the different feature sets.** File obtained with RF.
(CSV)

**S3 Data. Frequency of the top 25 features over 10 experiments for the 100% training set and the different feature sets.** File obtained with SVM.
(CSV)

## Acknowledgments

The authors would like to acknowledge the anonymous reviewers for their valuable revision of the manuscript.

Data used in the preparation of this article were obtained from the Alzheimer′s Disease Neuroimaging Initiative (ADNI) database (adni.loni.usc.edu). As such, the investigators within the ADNI contributed to the design and implementation of ADNI and/or provided data but did not participate in the analysis or writing of this report. A complete listing of ADNI investigators can be found at: http://adni.loni.usc.edu/wp-content/uploads/how_to_apply/ADNI_Acknowledgement_List.pdf.

## Author Contributions

**Conceptualization:** Monica Hernandez.

**Data curation:** Francisco Ferraz.

**Formal analysis:** Monica Hernandez.

**Funding acquisition:** Monica Hernandez.

**Investigation:** Monica Hernandez.

**Methodology:** Monica Hernandez, Ubaldo Ramon-Julvez, Francisco Ferraz.

**Software:** Monica Hernandez, Francisco Ferraz.

**Supervision:** Monica Hernandez.

**Writing – original draft:** Monica Hernandez.

**Writing – review & editing:** Ubaldo Ramon-Julvez.

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
