## [Decision Letter · Decision Letter 0]

12 Nov 2021

PONE-D-21-26983Explainable AI towards understanding the performance of the top-three Tadpole challenge methods in the forecast of Alzheimer's disease diagnosisPLOS ONE

Dear Dr. Hernandez,

Thank you for submitting your manuscript to PLOS ONE. After careful consideration, we feel that it has merit but does not fully meet PLOS ONE’s publication criteria as it currently stands. Therefore, we invite you to submit a revised version of the manuscript that addresses the points raised during the review process.

We look forward to receiving your revised manuscript.

Kind regards,

Khanh N.Q. Le

Academic Editor

PLOS ONE

Journal Requirements:

"Data collection and sharing for this project was funded by the Alzheimer's Disease 

Neuroimaging Initiative (ADNI) (National Institutes of Health Grant U01 AG024904)and DOD ADNI (Department of Defense award number W81XWH-12-2-0012). ADNI is

funded by the National Institute on Aging, the National Institute of Biomedical

Imaging and Bioengineering, and through generous contributions from the following: AbbVie, Alzheimer's Association; Alzheimer's Drug Discovery Foundation; Araclon 

Biotech; BioClinica, Inc.; Biogen; Bristol-Myers Squibb Company; CereSpir, Inc.;

Cogstate; Eisai Inc.; Elan Pharmaceuticals, Inc.; Eli Lilly and Company; EuroImmun; F.Hoffmann-La Roche Ltd and its affiliated company Genentech, Inc.; Fujirebio; GE 

Healthcare; IXICO Ltd.; Janssen Alzheimer Immunotherapy Research Development, 

LLC.; Johnson Johnson Pharmaceutical Research Development LLC.; Lumosity; 

Lundbeck; Merck Co., Inc.; Meso Scale Diagnostics, LLC.; NeuroRx Research;

Neurotrack Technologies; Novartis Pharmaceuticals Corporation; Pfizer Inc.; Piramal 

Imaging; Servier; Takeda Pharmaceutical Company; and Transition Therapeutics. The 

Canadian Institutes of Health Research is providing funds to support ADNI clinical

sites in Canada. Private sector contributions are facilitated by the Foundation for the National Institutes of Health (www.fnih.org). The grantee organization is the Northern California Institute for Research and Education, and the study is coordinated by the Alzheimer's Therapeutic Research Institute at the University of Southern California. ADNI data are disseminated by the Laboratory for Neuro Imaging at the University of Southern California. The authors would like to acknowledge the anonymous reviewers for their valuable revision of the manuscript. This work was partially supported by the national research grant PID2019-104358RB-I00 (DL-Aging project), and Government of Aragon Group Reference T64 20R (COS2MOS research group)"

"1) MH, URJ

Grant number: PID2019-104358RB-I00

Funder: Ministerio de Ciencia e Innovacion

URL: https://www.ciencia.gob.es/site-web/;jsessionid=FE84CCA1BA4EAA27EABCADF4007CCF07

2) MHG

Grant number: T 64 20R

Funder: Gobierno de Aragon

URL: https://www.aragon.es/

The funders had no role in study design, data collection and analysis, decision to publish, or preparation of the manuscript"

3. Please include your full ethics statement in the ‘Methods’ section of your manuscript file. In your statement, please include the full name of the IRB or ethics committee who approved or waived your study, as well as whether or not you obtained informed written or verbal consent. If consent was waived for your study, please include this information in your statement as well

Reviewers' comments:

Reviewer's Responses to Questions

**Comments to the Author**

1. Is the manuscript technically sound, and do the data support the conclusions?

Reviewer #1: Yes

Reviewer #2: Partly

Reviewer #3: Yes

Reviewer #4: Yes

2. Has the statistical analysis been performed appropriately and rigorously? 

Reviewer #1: Yes

Reviewer #2: No

Reviewer #3: Yes

Reviewer #4: Yes

3. Have the authors made all data underlying the findings in their manuscript fully available?

Reviewer #1: Yes

Reviewer #2: Yes

Reviewer #3: Yes

Reviewer #4: Yes

4. Is the manuscript presented in an intelligible fashion and written in standard English?

Reviewer #1: Yes

Reviewer #2: Yes

Reviewer #3: Yes

Reviewer #4: Yes

5. Review Comments to the Author

Reviewer #1: In this paper, the authors report on the TADPOLE (Alzheimer's Disease Prediction Of Longitudinal Evolution) project. The aim of this project is to identify the most predictive data, features and methods for the progression of patients at risk of developing Alzheimer's disease.

The challenge was successful in uncovering tree-based ensemble methods such as gradient boosting or random forests as the best methods for predicting the clinical state of Alzheimer's disease. However, the outcome of the competition was limited to which combination of data processing and methods had the best accuracy, making the contribution of methods to accuracy difficult to isolate. The quantification of feature importance was global of all methods used by the contest participants. In addition, Tadpole provided general answers aimed at improving performance, neglecting important aspects such as interpretability.

In this paper, the authors describe the models of the three best Tadpole methods in a common framework with the specific aim of providing a fair comparison. Furthermore, this paper also deals with explanations, in particular plausible explanations, why the methods have achieved this accuracy. For this purpose, the authors used well-known methods such as SHAP.

In this paper, the authors show that the two top-tadpole challenge methods are able to correctly determine those features most important for disease prognosis. This is good work, because it also reinforces confidence in such systems.

This reviewer finds this work important, relevant and interesting and would recommend acceptance and provides some recommendations for improvement below:

1a) Check the overall language, maybe let an English native speaker help to get a better flow, the content is good and can benefit from a nicer flow.

1b) Check the wording e.g. section 1, page 2 multi-modal -> both high-dimensional (many data points) and multi-modal (from different sources)

2) Table 1 is exceeding margins

3) Section 3.2, page 7 To make this paper up-to-date, authors could also mention a very recent work on actionable xAI with RF, see: https://arxiv.org/abs/2108.11674

4) Section 4, page 9, when introducing xAI the authors should also introduce the importantce of the quality of explanations particularly in a multi-modal setting, there is a very recent related work which can be helpful here, see: https://doi.org/10.1016/j.inffus.2021.01.008

5) Attention, Figure 1 is unreadable and exceeds margin

6) Figures 2 to 4 are diffiult to read - maybe enlarge?

7) All the further figures the same- the reviewer does not know how the solution could be here ? But take care that the reader is not confronted with difficult readability.

Reviewer #2: Unfortunately, explainable AI means a more general concept. XAI should explain why technical findings contribute to medical findings. This reviewer cannot see any discussion on the biological aspects of AD derived from methods but a mere technical discussion.

Reviewer #3: This is a well-designed and well-executed study of the three top models in the TADPOLE challenge to predict evolution in Alzheimer's disease with machine learning models.

Importantly, they have confirmed the ability of the top two models in the challenge to predict outcome. As part of the Explainable AI initiative they have identified the most important features that predict progression in Alzheimer disease.

This is important work since many proposed models for prognostication never get validated independently on the original dataset.

1. I would defer to the editor and authors, but it would seem that TADPOLE like SHAP is an acronym and should be capitalized throughout.

2. The word "antagonist" on line 35 (page 2) seems wrongly used. Seems like they mean "Clinical research to mitigate Alzheimer's disease has taken two different directions."

3.The Glossary on pages 1-2 is very helpful. Perhaps a few more abbreviations from Figs 2-10 can be added.

4. The interpretation of the Feature labels from Figs 2-10 is difficult--requiring reader to refer to the initial data dictionary on line. I suggest this is an issue for the Editor and Authors to resolve.

5. The font on figures 2-10 are barely legible. Again I suggest that the Editor and Authors resolve this issue.

6. Verifying is misspelled on line 878. I hope that authors will spell check entire manuscript.

7. Given the length of this manuscript, I am not sure the Titanic figure and discussion adds anything of value (Figure 1).

8. Color coding some of the bars of Figures 3, 4, etc. by feature category (e.g. Cognitive, Imaging, Biomarker, etc) may highlight differences in feature selection more dramatically.

Reviewer #4: The manuscript outlines a timely and important piece of research. The study is well contextualised in the coverage of the literature, and the need for explainable AI in medical imaging is well justified. The sections that follow provide a logical account of the work conducted, including useful discussions of the data, the preprocessing methods, the predictive models and the explainable AI methods used. Enough detail is provided to allow other researchers to replicate the process. The results are presented along with a thoughtful exploration of the importance of feature set selection and sample size. The subsequent analysis of interpretability is comprehensive, and supports the concluding remarks about the performance of the models, the most meaningful features, and the consistency between the models and clinical knowledge. In essence, this is a robust piece of work, but it is sometimes undermined by the presentation.

There are a number of flaws which should be addressed:

1). The authors should provide details on the analysis/justification of the validation strategy. A train/test/evaluation split is conducted – might k-fold cross-validation be considered in addition or as an alternative? Why was k-fold cross validation not used?

2). The authors should acknowledge the weakness in their validation. The test and evaluation data originate from the same study and distribution, and therefore the performance is unlikely to reflect the models’ performance in a truly external dataset or clinical setting.

3). Intelligibility and standard of English. This is the area the manuscript has the most flaws: there are frequent grammar and spelling errors which obfuscate the meaning. These are really too numerous to list. The manuscript should be very carefully proofread by a native English speaker for errors to improve readability.

4). The plots are good, but there are many of them which means that they and their labels are small and condensed, making them difficult to inspect and compare. The authors should enlarge the plots. This may require rearrangement, being more selective about plot inclusion, or some sort of aggregation of plots.

6. PLOS authors have the option to publish the peer review history of their article (what does this mean?). If published, this will include your full peer review and any attached files.

Reviewer #1: No

Reviewer #2: No

Reviewer #3: **Yes: **None

Reviewer #4: No

---

## [Author Response · Author response to Decision Letter 0]

14 Jan 2022

Response to the reviewers

We want to thank the reviewers for their work in the review process. We believe that their thorough comments helped us to improve the quality of the manuscript. We have revised the paper in accordance with their suggestions to the best of our ability. We tried to answer all the concerns raised by the reviewers. We made a particular effort in improving the English style of the manuscript by using Editage services. We also made an effort on improving the readability of the figures. We hope that the revised version of the manuscript will now meet the standards required for publication.

We would like to thank Reviewer 1 pointer to the recent state-of-the-art in explainable and actionable AI. In addition, we believe that the color-coding idea given by Reviewer 3 helped improve the readability of the figures. We would like to thank Reviewer 4 for his/her suggestion on our study limitation.

In the following pages, we include a point-by-point summary of our responses to the comments. Reviewer’s points are presented in italics, while our responses are presented in normal text after “Response:”. In the modified version of the manuscript, the corrected text is colored in blue.

Reviewer #1:

Summary

-----------

In this paper, the authors report on the TADPOLE (Alzheimer's Disease Prediction Of Longitudinal Evolution) project. The aim of this project is to identify the most predictive data, features and methods for the progression of patients at risk of developing Alzheimer's disease.

The challenge was successful in uncovering tree-based ensemble methods such as gradient boosting or random forests as the best methods for predicting the clinical state of Alzheimer's disease. However, the outcome of the competition was limited to which combination of data processing and methods had the best accuracy, making the contribution of methods to accuracy difficult to isolate. The quantification of feature importance was global of all methods used by the contest participants. In addition, Tadpole provided general answers aimed at improving performance, neglecting important aspects such as interpretability.

In this paper, the authors describe the models of the three best Tadpole methods in a common framework with the specific aim of providing a fair comparison. Furthermore, this paper also deals with explanations, in particular plausible explanations, why the methods have achieved this accuracy. For this purpose, the authors used well-known methods such as SHAP.

In this paper, the authors show that the two top-tadpole challenge methods are able to correctly determine those features most important for disease prognosis. This is good work, because it also reinforces confidence in such systems.

This reviewer finds this work important, relevant and interesting and would recommend acceptance and provides some recommendations for improvement below:

Main Points

-------------

1a) Check the overall language, maybe let an English native speaker help to get a better flow, the content is good and can benefit from a nicer flow.

Response: We hired Editage services to improve the overall language (https://www.editage.com/). Editage’s editor provided a complete revision of the paper and we translated all his suggestions to the manuscript. We expect that our work can now be better followed up without English mistakes. Since the editing was exhaustive we did not color these changes. 

1b) Check the wording e.g. section 1, page 2 multi-modal -> both high-dimensional (many data points) and multi-modal (from different sources)

Response: We included the reviewer’s suggestion in the manuscript. Thank you very much for improving the meaning of the sentence.

1.2) Table 1 is exceeding margins

Response: The LaTeX template from PLOS One produces the page layout shown in Figure 1.2.1. The tables are apparently allowed to exceed margins in the LaTeX documentclass article. The page layout of our manuscript including Table 1 is shown in Figure 1.2.2.

Figure 1.2.1. Layout of PLOS One LaTeX template including a table.

Figure 1.2.2. Layout of our manuscript including Table 1.

1.3) Section 3.2, page 7 To make this paper up-to-date, authors could also mention a very recent work on actionable xAI with RF, see: https://arxiv.org/abs/2108.11674

Response: We included a sentence regarding the missing link between explainable AI (the methods cited in the paper) and actionable explainable AI (the citation suggested by the reviewer) and a comment on alternatives for dealing with multi-modal information provided in the citation suggested by the reviewer. We consider that the changes should be performed in the Introduction instead of Section 3.2. 

1.4) Section 4, page 9, when introducing xAI the authors should also introduce the importance of the quality of explanations particularly in a multi-modal setting, there is a very recent related work which can be helpful here, see: https://doi.org/10.1016/j.inffus.2021.01.008

Response: We would like to thank the reviewer for this interesting piece of work linking explainability with trustable and actionable systems for aiding decision-making in clinical applications. We included in Section 4 a paragraph condensing the ideas provided in that citation. We also included some other citations to related works on causability. 

1.5) Attention, Figure 1 is unreadable and exceeds margin

Response: We changed the subfigure organization from horizontal to vertical and reduced the size of the figures to respect margins.

1.6) Figures 2 to 4 are difficult to read - maybe enlarge?

Response: We have transposed the layout of Figures 2 to 4. This way we have been able to enlarge the figures by 10% of the textwidth.

1.7) All the further figures the same- the reviewer does not know how the solution could be here ? But take care that the reader is not confronted with difficult readability.

Response: We have changed the layout of Figure 5 from a horizontal to a vertical arrangement. This way we have been able to enlarge the figures by 10% of the textwidth. We could not further enlarge since we are limited by the textheight also.

For the remaining figures (Figure 6 to 10), we were not able to find a satisfactory solution. We first considered further trimming the labels. However, some substrings of important labels were removed hindering a correct identification of the label (e.g. different RAVLT scores). We also considered a layout arrangement change, as it worked in Figures 2 to 4. However, the textheight limitation forced the use of exactly the same figure size as originally. Finally, we considered the rearrangement and combination of different Figures. However, the textheight limitation also forced us to keep the original figure size. 

For Figures 6 and 7 we could not find any better solution than reading the digital version of the manuscript with zooming. Labels can be perfectly read with a zoom of 300%. We hope this would be enough for the reader to be able to analyze the interesting information.

For Figures 8, 9, and 10 we have ordered the features with the same frequency by alphabetical order and we have included the frequency results in the csv files of the supplementary material: XGB_Summary_Plot_Robustness.csv, RF_Summary_Plot_Robustness.csv, and SVM_Summary_Plot _Robustness.csv, respectively. Thus, the interested reader can directly analyze the information regarding the frequency of features in the top 25.

Finally, we used the Preflight Analysis and Conversion Engine (PACE) digital diagnostic tool, https://pacev2.apexcovantage.com/ provided by PLOSOne in order to ensure that the figures meet PLOS requirements.

Reviewer #2:

Unfortunately, explainable AI means a more general concept. XAI should explain why technical findings contribute to medical findings. This reviewer cannot see any discussion on the biological aspects of AD derived from methods but a mere technical discussion.

Response: We recall the definitions found in the article kindly pointed out by reviewer 1 https://doi.org/10.1016/j.inffus.2021.01.008

Explainability ∶= technically highlights decision relevant parts of machine representations and machine models i.e., parts which contributed to model accuracy in training, or to a specific prediction for a particular observation. Here the xAI community has already developed a variety of successful methods. Explainability does not refer to a human model.

 Causality ∶= the relationship between cause and effect in the sense of Judea Pearl.

Causability ∶= the measurable extent to which an explanation – resulting from (A) – to a human expert achieves a specified level of causal understanding. This can be measured e.g. with the System Causability Scale. Causability refers to a human model.

According to the first definition, we believe that our use of explainability is correct. We focused on the importance given to the features by the different models and verified whether the features arising in the top list are consistent with clinical knowledge. The way of using XAI methods was technical, no human model was involved in the process. Of course, our contribution is a first step towards actionable explainable methods. With our approach, we can only question the reliability of SVM, since the model strongly changes the importance given to different features among different bootstraps. In addition, we find that random forests mostly rely on cognitive features and this may be a problem for models aiming at early diagnosis. 

Since the 1818 features are related to the biological aspects of AD we could only provide a ranking of features according to their importance in model decisions. In a setting with features related and not related to the biological aspects of AD or unknown, our proposed methodology may be used to assess whether the best performing models are able to ignore non-important features or give credit to some unknown feature to the date. This information would result in an increase or decrease of trust in the systems and features. 

The connection of the biological aspects of AD and our technical explanations would fit into causability and actionability, where, with the help of a clinical specialist, we would evaluate the quality of our explanations and establish whether the explanations of the model enable some causal understanding of the disease. This could be a huge step towards computer- aided diagnosis of AD but it is outside the scope of the present work. 

Reviewer #3:

This is a well-designed and well-executed study of the three top models in the TADPOLE challenge to predict evolution in Alzheimer's disease with machine learning models.

Importantly, they have confirmed the ability of the top two models in the challenge to predict outcome. As part of the Explainable AI initiative they have identified the most important features that predict progression in Alzheimer disease.

This is important work since many proposed models for prognostication never get validated independently on the original dataset.

3.1. I would defer to the editor and authors, but it would seem that TADPOLE like SHAP is an acronym and should be capitalized throughout.

Response: We have replaced all appearances of Tadpole with TADPOLE in the manuscript. In addition, we have checked that SHAP is written in upper case all over the manuscript.

3.2. The word "antagonist" on line 35 (page 2) seems wrongly used. Seems like they mean "Clinical research to mitigate Alzheimer's disease has taken two different directions."

Response: We are sorry for our use of English in this sentence. The word antagonist makes perfect sense in the authors’ mother tongue. We replaced the word antagonist with the author’s suggestion. 

3.3. The Glossary on pages 1-2 is very helpful. Perhaps a few more abbreviations from Figs 2-10 can be added.

Response: We followed the reviewer’s suggestion and added to the glossary some of the most frequent features appearing in Figures 2 - 10. Thank you for the suggestion, we believe that the reader will find the list of terms very useful. 

3.4. The interpretation of the Feature labels from Figs 2-10 is difficult--requiring readers to refer to the initial data dictionary online. I suggest this is an issue for the Editor and Authors to resolve.

Response: Actually, the interpretation of the feature labels can be found through the text of Section 5. We have focused on the labels related to the most relevant results. With the name of the corresponding label, we include the explanation of the kind of label and finer details. For example, in the paragraph

“The most relevant features include the diagnosis, the age of each patient at baseline (Years bl), and a number of measurements from cognitive tests (CDRSB, FAQ, ADAS13, RAVLT, CDRSB bl, DX bl, ADAS11, and ECog). Then, a number of measurements from anatomical structures obtained from FreeSurfer cross-sectional pipeline (UCSFFSX) arise, interleaved with ADNIMERGE image and other cognitive measurements. The anatomical structures include the amygdala (ST17SV), the hippocampus (ST29SV), the inferior and middle temporal regions (ST32TA, ST40TA), the entorhinal region (ST83CV), and the cuneus (ST23SA).”

we first mentioned labels from cognitive tests and then labels from MRI cross-sectional biomarkers. In addition, we detailed the structures associated with the biomarkers since they are relevant to the diagnosis of Alzheimer’s Disease. The reader can complete the interpretation of the labels with the information given in the completed glossary or the information in the referred webpage https://github.com/swhustla/pycon2017-alzheimers-hack/blob/master/docs/data_dictionary.md.

3.5. The font on figures 2-10 are barely legible. Again I suggest that the Editor and Authors resolve this issue.

Response: We tried to solve this issue to the best of our ability. Please, see our response to Reviewer 1 comments on points 1.6 and 1.7.

3.6. Verifying is misspelled on line 878. I hope that authors will spell check the entire manuscript.

Response: We hired Editage services to improve the overall language (https://www.editage.com/). Editage’s editor provided a complete revision of the paper and we translated all his suggestions to the manuscript. We expect that our work can now be better followed up without English mistakes. Since the editing was exhaustive we did not color these changes. 

3.7. Given the length of this manuscript, I am not sure the Titanic figure and discussion adds anything of value (Figure 1).

Response: In order to reduce the length of the manuscript, we moved the enlarged Titanic figure to the supplementary material. We believe that Figure 1 in the context of Titanic survivors is a good example for explaining how to interpret the information given in SHAP figures in a use case that is accessible even to non-specialists. We believe readers not used to SHAP may find it useful to understand first the interpretation of SHAP figures in a much simpler example and then move to understand the interpretation of SHAP figures in the context of Alzheimer’s Disease.

3.8. Color coding some of the bars of Figures 3, 4, etc. by feature category (e.g. Cognitive, Imaging, Biomarker, etc) may highlight differences in feature selection more dramatically.

Response: Apart from the legibility improvement in response to point 1.7, we have used a color-coding for the bars of Figures 2, 3, 4, 8, 9, 10. We believe that this is a great idea that has improved the legibility of the plots and we thank the reviewer for the suggestion.

Reviewer #4:

The manuscript outlines a timely and important piece of research. The study is well contextualised in the coverage of the literature, and the need for explainable AI in medical imaging is well justified. The sections that follow provide a logical account of the work conducted, including useful discussions of the data, the preprocessing methods, the predictive models and the explainable AI methods used. Enough detail is provided to allow other researchers to replicate the process. The results are presented along with a thoughtful exploration of the importance of feature set selection and sample size. The subsequent analysis of interpretability is comprehensive, and supports the concluding remarks about the performance of the models, the most meaningful features, and the consistency between the models and clinical knowledge. In essence, this is a robust piece of work, but it is sometimes undermined by the presentation.

There are a number of flaws which should be addressed:

4.1). The authors should provide details on the analysis/justification of the validation strategy. A train/test/evaluation split is conducted – might k-fold cross-validation be considered in addition or as an alternative? Why was k-fold cross validation not used?

Response: In fact, k-fold cross-validation with 5 folds was used in the selection of the hyperparameters through the class RandomizedSearchCV

https://scikit-learn.org/stable/modules/generated/sklearn.model_selection.RandomizedSearchCV.html. We have included this detail in the description of the TADPOLE challenge methods (Section 3).

Regarding our validation strategy, it was the one used in the TADPOLE challenge. The rationale behind this choice is to show that our models are able to obtain test and evaluation results similar to Frog, ThreeDays, and EMC-EB models while meeting the objective of isolating the contribution of pure machine-learning methods to the accuracy in diagnosis.

4.2). The authors should acknowledge the weakness in their validation. The test and evaluation data originate from the same study and distribution, and therefore the performance is unlikely to reflect the models’ performance in a truly external dataset or clinical setting.

Response: We do agree with the reviewer that the evaluation and interpretability results of our models may not be preserved with different test and evaluation datasets. We believe that a very interesting direction would be to corroborate our findings with external clinical datasets. Towards this end, we are currently working with Dr. Antonio Lobo, from the Lozano Blesa University Hospital in our town Zaragoza in order to apply our methods to an in-house database of psychiatric records (ZARADEMP project, https://cheba.unsw.edu.au/consortia/cosmic/studies/zarademp-project).

We added a paragraph in the Discussion and Conclusions section of our manuscript acknowledging the limitation. 

4.3). Intelligibility and standard of English. This is the area the manuscript has the most flaws: there are frequent grammar and spelling errors which obfuscate the meaning. These are really too numerous to list. The manuscript should be very carefully proofread by a native English speaker for errors to improve readability.

Response: We hired Editage services to improve the overall language (https://www.editage.com/). Editage’s editor provided a complete revision of the paper and we translated all his suggestions to the manuscript. We expect that our work can now be better followed up without English mistakes. Since the editing was exhaustive we did not color these changes. 

4.4). The plots are good, but there are many of them which means that they and their labels are small and condensed, making them difficult to inspect and compare. The authors should enlarge the plots. This may require rearrangement, being more selective about plot inclusion, or some sort of aggregation of plots.

Response: We tried to solve this issue to the best of our ability. Please, see our response to Reviewer 1 comments on points 1.6 and 1.7.

---

## [Decision Letter · Decision Letter 1]

16 Feb 2022

Explainable AI toward understanding the performance of the top-three Tadpole challenge methods in the forecast of Alzheimer's disease diagnosis

PONE-D-21-26983R1

Dear Dr. Hernandez,

We’re pleased to inform you that your manuscript has been judged scientifically suitable for publication and will be formally accepted for publication once it meets all outstanding technical requirements.

Kind regards,

Nguyen Quoc Khanh Le

Academic Editor

PLOS ONE

Additional Editor Comments (optional):

Reviewers' comments:

Reviewer's Responses to Questions

**Comments to the Author**

1. If the authors have adequately addressed your comments raised in a previous round of review and you feel that this manuscript is now acceptable for publication, you may indicate that here to bypass the “Comments to the Author” section, enter your conflict of interest statement in the “Confidential to Editor” section, and submit your "Accept" recommendation.

Reviewer #1: All comments have been addressed

Reviewer #4: All comments have been addressed

2. Is the manuscript technically sound, and do the data support the conclusions?

Reviewer #1: Yes

Reviewer #4: Yes

3. Has the statistical analysis been performed appropriately and rigorously? 

Reviewer #1: Yes

Reviewer #4: Yes

4. Have the authors made all data underlying the findings in their manuscript fully available?

Reviewer #1: Yes

Reviewer #4: No

5. Is the manuscript presented in an intelligible fashion and written in standard English?

Reviewer #1: Yes

Reviewer #4: Yes

6. Review Comments to the Author

Reviewer #1: The paper has much improved and this reviewer is now arguing for accepting this paper, of course the authors should do the usual final spell checks and language checks - but ocntent wise the paper is now fine and of value for the interested reader.

Reviewer #4: The authors have addressed all my concerns in their revision. I am happy to recommend acceptance.

I have no further concerns.

7. PLOS authors have the option to publish the peer review history of their article (what does this mean?). If published, this will include your full peer review and any attached files.

Reviewer #1: No

Reviewer #4: No

---

## [Editor Report · Acceptance letter]

28 Apr 2022

PONE-D-21-26983R1 

Explainable AI toward understanding the performance of the top three TADPOLE Challenge methods in the forecast of Alzheimer's disease diagnosis 

Dear Dr. Hernandez:

I'm pleased to inform you that your manuscript has been deemed suitable for publication in PLOS ONE. Congratulations! Your manuscript is now with our production department. 

Kind regards, 

on behalf of

Dr. Nguyen Quoc Khanh Le 

Academic Editor

PLOS ONE